# A constricted mitochondrial morphology formed during respiration

Manish K. Singh [1,2], Laetitia Cavellini[1,7], Maria Angeles Morcillo-Parra[1,7], Christina Kunz[3], Mickaël Lelek[2], Perrine Bomme [4], Aurélia Barascu[1], Cynthia Alsayyah[1,6], Maria Teresa Teixeira[1], Naïma Belgareh-Touzé[1], Adeline Mallet [4], Lea Dietrich[3], Christophe Zimmer [2,5] & Mickael M. Cohen [1] ✉

Mitochondria assemble in a dynamic tubular network. Their morphology is governed by mitochondrial fusion and fission, which regulate most mitochondrial functions including oxidative phosphorylation. Yet, the link between mitochondrial morphology and respiratgion remains unclear. Here, we uncover a mitochondrial morphology dedicated to respiratory growth of *Saccharomyces cerevisiae*, which we refer to as "Ringo". The Ringo morphology is characterized by stable constrictions of mitochondrial tubules. Ringo constrictions are mediated by the yeast dynamin Dnm1 and, unlike mitochondrial fission, occur in the absence of contacts with the endoplasmic reticulum. Our data show that blocking formation of the Ringo morphology correlates with decreased respiration, decreased expression of OXPHOS subunits and perturbed mitochondrial DNA distribution. These results open important perspectives about the link between mitochondrial form and function.

Mitochondria form tubular structures inside the cell that associate in an intricate membrane network. The morphology of these tubules is conditioned by an equilibrium between frequent fusion and fission events mediated by large GTPases of the Dynamin-Related Proteins (DRP) superfamily[1–3]. Among the DRPs, Dnm1 and DRP1 mediate fission of the outer mitochondrial membrane in yeast and metazoans, respectively[2,4–6]. The DRPs are recruited to mitochondrial membranes by specific mitochondrial adaptors that accumulate at sites of contact between mitochondria and the Endoplasmic Reticulum (ER). Once recruited, Dnm1 and DRP1 auto-oligomerize in a GTP-dependent manner to form macromolecular spirals which wrap around mitochondrial tubules that have been pre-constricted by the ER[6–9]. GTP hydrolysis reduces the diameter of the spiral, resulting in hyper-constriction of the mitochondrial tubule and eventual fission. These sites of mitochondrial

fission in contact with the ER are established as platforms of mitochondrial DNA (mtDNA) replication[10,11]. Together, mitochondrial fusion and fission guide mitochondrial dynamics and regulate key mitochondrial functions including oxidative phosphorylation[1,12,13]. Cristae density increases in respiratory conditions and is associated with increased respiratory function as well as accumulation of mtDNA[14–17]. Since the ground-breaking electron microscopy observation that mitochondria in yeast are one, giant branched cellular organelle[18], fluorescence microscopy and continued improvement in resolution provided by confocal imaging revealed that mitochondrial networks are more branched with thicker tubules in respiratory as compared to non-respiratory conditions[13]. Other studies in *Saccharomyces cerevisiae* or *pombe* proposed even more dramatic remodelling of the mitochondrial network with extensive fragmentation during respiration[12,19,20]. With the initial

[1]Laboratoire de Biologie Moléculaire et Cellulaire des Eucaryotes, Sorbonne Université, CNRS, UMR8226, Institut de Biologie Physico-Chimique, Paris, France. [2]Institut Pasteur, Université Paris Cité, CNRS UMR 3691, Imaging and Modeling Unit, Paris, France. [3]Max Planck Institute of Biophysics, Department of Structural Biology, Frankfurt am Main, Germany. [4]Ultrastructural BioImaging Core Facility, C2RT, Institut Pasteur, Université Paris Cité, Paris, France. [5]Rudolf Virchow Center for Integrative and Translational Bioimaging, University of Würzburg, Würzburg, Germany. [6]Present address: Medical Faculty, Medical Biochemistry and Molecular Biology, Saarland University, Homburg, Germany. [7]These authors contributed equally: Laetitia Cavellini, Maria Angeles Morcillo-Parra. ✉e-mail: cohen@ibpc.fr

objective of understanding this discrepancy, we analysed respiratory mitochondrial morphologies further using super-resolution imaging.

## Results

### Visualization of the Ringo mitochondrial morphology

The model organism *Saccharomyces cerevisiae* is an ideal system to study the crosstalk between mitochondrial dynamics and metabolism. Depending on the carbon source provided in the culture media, *S. cerevisiae* can grow either through respiration or fermentation. Glycerol or ethanol induces respiratory growth, whereas D-glucose, also called dextrose, inhibits respiration even in the presence of oxygen, and is a preferred fermentative carbon sources[21,22].

To decipher whether mitochondrial networks are more branched or more fragmented during respiratory growth, we analysed mitochondrial morphology during fermentation (dextrose containing media, YPD) or respiration (glycerol containing media, YPG) first using spinning disk microscopy in cells in which the outer mitochondrial membrane (Tom70-GFP) and matrix (mito-mCherry) were fluorescently labelled. In spinning disk images, whose lateral resolution is > 200 nm, mitochondrial outer membranes and matrices were co-localized and no differences in morphology were observed during fermentation compared with respiration (Fig. 1a). Structured Illumination Microscopy (SIM) is a super-resolution imaging method that improves the resolution to ~125 nm. Thanks to the improved resolution of SIM, mitochondria in fermentation conditions clearly appeared as tubes, with a thin tubular matrix surrounded by the outer membrane (Fig. 1b and Supplementary Fig. 1a). Strikingly, SIM revealed that tubules underwent extensive rearrangements upon respiration. This reorganization is reflected in numerous connexions of the outer membrane, which resemble beads on a string in 2D z-projection. The matrix, on the other hand, appeared disconnected (Fig. 1b and Supplementary Fig. 1a). 3D reconstitution of the outer membrane confirmed its unexpected organization upon respiratory growth (Supplementary Movie 1). We chose to name this unknown morphology as "Ringo".

To verify this finding with a different imaging method, we used photo activated localization microscopy (PALM), which achieves spatial resolutions of 20-50 nm, in Tom70-mEos2 cells. Mitochondrial networks imaged by PALM in cells grown in dextrose (YPD) were tubular, while those imaged in cells grown in glycerol (YPG) were formed of connected rings similar to those captured by SIM, providing independent confirmation of the Ringo morphology (Fig. 1c and Supplementary Fig. 1b). The Ringo morphology was observed regardless of the protein used to label the outer mitochondrial membrane with either Om45-GFP (Supplementary Fig. 1c) or Tom70-GFP (Fig. 1b and Supplementary Movie 1), indicating that this phenotype is a property of the outer membrane shape rather than of specific membrane proteins.

### The Ringo mitochondrial morphology is linked to respiratory growth

Our clear-cut observation that the Ringo morphology occurs during cell growth in glycerol-containing media, while the mitochondrial network is essentially tubular in the presence of dextrose, suggests a possible link between the formation of Ringo mitochondrial networks and respiratory growth. To further test this association, we analysed mitochondrial morphology in the presence of ethanol, another carbon source that promotes respiration. Again, the Ringo morphology was dominant (Supplementary Fig. 2a). When yeast is grown in dextrose media, this source of carbon is progressively depleted, and cells switch to self-produced ethanol as a respiratory carbon source to continue growing. If the Ringo morphology is associated with respiration, we therefore expect a switch from a tubular morphology to the Ringo morphology as function of time. We indeed observe this switch after 4–6 h of growth in 2% dextrose media (Supplementary Fig. 2b). If this

switch is due to dextrose depletion, it should occur earlier if less dextrose is available initially as seen in Supplementary Fig. 2c. Conversely, we confirmed that the Ringo morphology reverts to the classical tubular morphology when cells get transferred from glycerol to dextrose media (Supplementary Fig. 2d). Unlike in glycerol or ethanol, cells grown in galactose or raffinose media can perform both respiratory and fermentative growth[23–25]. If Ringo and tubular morphologies are associated with respiration and fermentation, respectively, we expect a mixture of these morphologies. Indeed, we observed that when grown in media with either of these two distinct carbon sources, 27–29% of cells displayed tubular mitochondria whereas 60–63% had Ringo mitochondrial networks (Supplementary Fig. 2e). These experiments further support a close correlation between the Ringo morphology and respiration.

### The Ringo morphology is formed by constrictions of the mitochondrial network

Axial projections of cells expressing Tom70-GFP (outer membrane) and mito-mCherry (matrix) as mitochondrial markers of tubular (Supplementary Fig. 3a), fragmented (Supplementary Fig. 3b) or Ringo networks (Supplementary Fig. 3c), suggested that mitochondria with Ringo morphology are not separated from each other (Supplementary Fig. 3c). Instead, the matrix seems to remain continuous throughout the Ringo network (Supplementary Fig. 3c), suggesting an unconventional mitochondrial architecture maintained by repetitive constrictions of mitochondrial tubules. However, the resolution of SIM or PALM are insufficient to totally rule out discontinuous membranes in close proximity resulting from a fission event. We therefore complemented fluorescence imaging with Transmission Electron microscopy (TEM) and Cryo-Electron Tomography (Cryo-ET) analysis. These methods confirmed that mitochondria of cells grown in dextrose-containing media (YPD) were overall tubular, whereas mitochondria of cells grown in glycerol-containing media (YPG) were remodelled by numerous constrictions (Fig. 1d, e and Supplementary Figs. 4a–c), approximately four times more frequent in one μm length of Ringo than in one μm length of tubular networks (Supplementary Fig. 4b). The diameter of Ringo mitochondria (0.31–0.44 μm) was wider than that of tubular mitochondria (0.23–0.35 μm) (Fig. 1f). This is in full agreement with diameters of fermentative and respiratory mitochondria measured previously[13]. Ringo networks were also significantly longer and more branched than tubular networks (Supplementary Fig. 5), which is consistent with previous observations[13]. In glycerol as compared to dextrose media, the number of mitochondrial constrictions (Fig. 1g) increased by 3 (Cryo-ET) to 6 (SIM and TEM) fold and the number of rings (Fig. 1h) by 2 (TEM), 3 (cryo-ET) or 6 (SIM) fold. These results thus confirm major changes in mitochondrial morphology upon respiratory (YPG) as compared to fermentative growth (YPD) and reveal that the Ringo morphology is formed by numerous constrictions of mitochondrial tubules.

### Dnm1, Mdv1 and Fis1 are required for formation of the Ringo morphology

Mitochondrial constriction is generally considered to be a prerequisite for fission, which in yeast involves Dnm1[2,4–6]. If Dnm1 also promotes constrictions in the Ringo morphology, its ablation should affect formation of this phenotype. SIM analysis of *dnm1Δ* cells labelled with Tom70-GFP and mito-mCherry confirmed that the absence of Dnm1 results in hyperfused mitochondria[4] upon fermentative growth (Fig. 2a, Supplementary Fig. 6a and Supplementary Movie 2, YPD). Respiratory conditions that produce the Ringo morphology in *WT* cells, led to another previously unknown mitochondrial phenotype in *dnm1Δ* cells (Fig. 2a, Supplementary Fig. 6a and Supplementary Movie 2, YPG). The respiratory *dnm1Δ* mitochondrial network is characterized by significantly wider mitochondrial diameters and equally frequent but obviously more random constrictions than in the

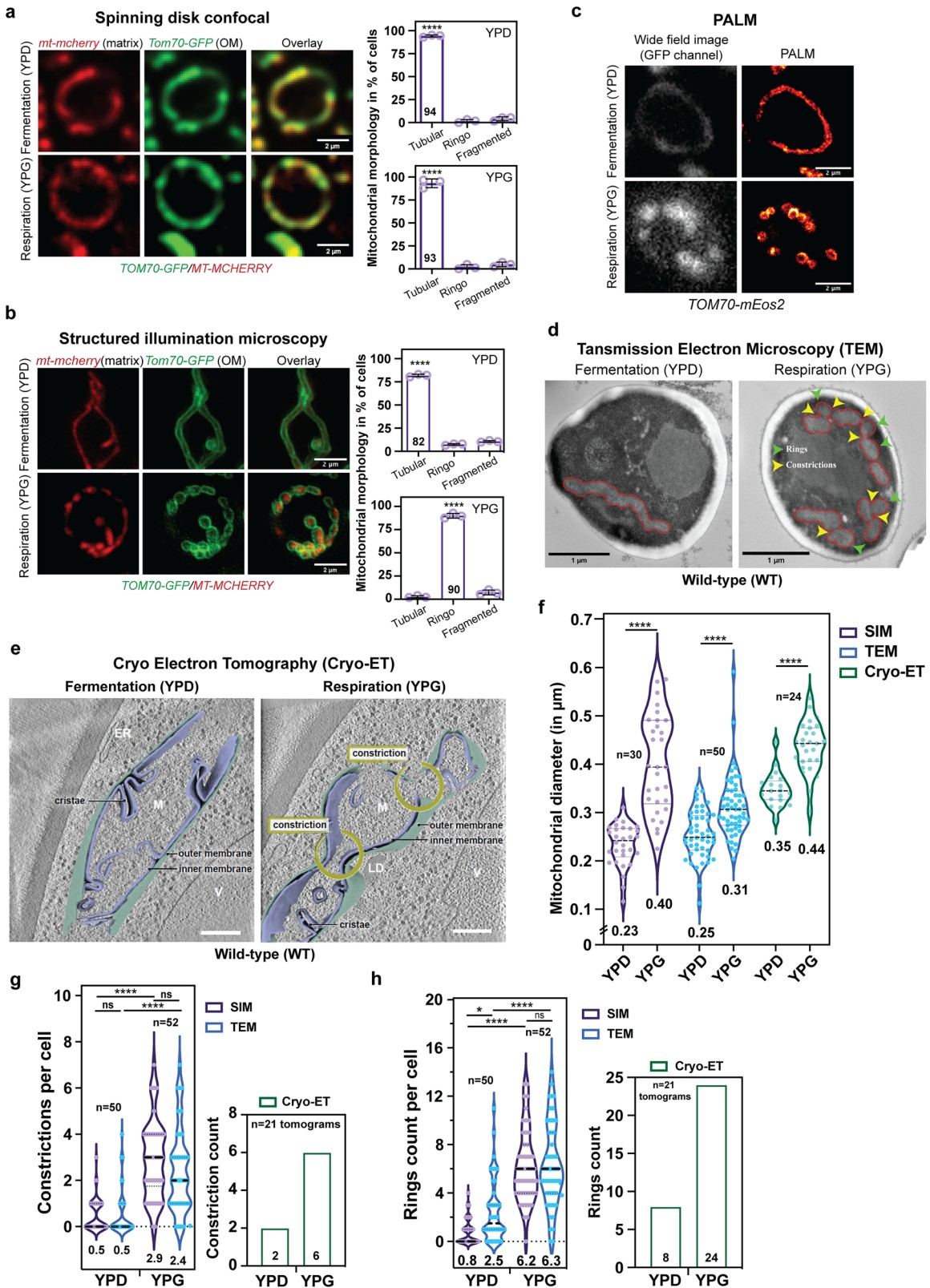

Ringo morphology (Supplementary Fig. 6b; compare Supplementary Movies 1 and 2). This new mitochondrial morphology can be considered as a HyperFused Respiratory (HFR) mitochondrial network and will, as of now, be named HFR.

Mitochondrial recruitment of Dnm1 occurs through the mitochondrial adaptors Mdv1 or Caf4 which can themselves interact with the mitochondrial anchor Fis1[26–31]. Absence of Caf4 neither affected

tubular morphology in fermentation nor the Ringo morphology in respiration (Fig. 2b and Supplementary Fig. 6c). In *mdv1Δ* and *fis1Δ* cells, however, the Ringo morphology was no longer observed in respiratory growth and was instead replaced by the HFR morphology (Fig. 2c–d and Supplementary Fig. 6d–e). These results indicate that proper formation of the Ringo phenotype requires Dnm1, Mdv1 and Fis1 but not Caf4.

**Fig. 1 | Identification and characterization of the Ringo mitochondrial morphology. a, b** Spinning-disc acquisitions (**a**) or SIM z-projections (**b**) of cells labeled for mitochondrial matrix (mt-mCherry) and Outer Membranes (Tom70-GFP) in fermentation (top) or respiration (bottom). Scale bar, 2 μm. Right graphs: percentage of cells with Tubular, Ringo or Fragmented mitochondria. Mean ± s.d. from > 41 cells (**a**) or > 86 cells (**b**) per $n = 3$ independent experiments (purple circles). ****$p < 0.0001$ (One-way Anova test followed by Tukey's multiple comparisons test) (**c**) Representative wide-field or PALM images of cells labeled with TOM70-mEOS2 in fermentation or respiration (see also Supplementary Fig. 1b with images obtained from two independent experiments). **d** Representative TEM micrographs (Scale bar, 1 μm) of cells in fermentation or respiration (see also Supplementary Fig. 4a with images obtained from two independent experiments). Mitochondria are delimited by red demarcations. Rings and constrictions are indicated by green or yellow arrowheads, respectively. **e** Slices through tomographic volumes and 3D renderings of mitochondria (M) during fermentation or respiration. Vacuoles (V), Lipid Droplets (LD) and mitochondrial sub-compartments are indicated. Scale bars, 500 nm. **f** Average mitochondrial diameter per cell, ****$p < 0.0001$ (**g**) constriction counts per cell (left) or per Tomogram (right), ****$p < 0.0001$, ns $p = 0.9989$ or 0.2978 and (**h**) ring counts per cell (left) or per Tomogram (right), in YPD or YPG as quantified in SIM, TEM, or Cryo-ET, *$p = 0.0107$, ****$p < 0.0001$, ns $p = 0.9991$. In **f**–**h**, violin plots from $n = 24$ to 50 cells (**f**) or $n = 50$ to 52 cells (**g, h**) with mean indicated at the bottom and by black dashed lines (quartiles with dotted lines). In **g**, **h**, $n = 21$ tomograms (Cryo-ET). In **f**–**h**, ns, not significant (Two-way Anova test followed by Tukey's multiple comparisons test).

Cryo-ET analysis confirmed the established cristae increase in *WT* cells during respiration *vs* fermentation (Fig. 2f)[32,33]. While abrogation of Dnm1 expression blocks formation of Ringo networks, it does not block the increase in Cristae density between fermentation and respiration (Fig. 2f). This suggests that the increase in cristae density and the constrictions of outer membranes within Ringo networks are promoted by parallel but distinct mechanisms. Cryo-ET analysis further revealed that cristae from *dnm1Δ* cells grown in respiratory conditions display a significant reorganization as compared to cristae from *WT* cells (Fig. 2f, g). Cristea of HFR mitochondria were not only much longer than those of Ringo networks (Fig. 2g) but also often extended across the whole width of the organelle (Fig. 2h), leading to several separations of the mitochondrial matrix (Fig. 2e, bridging cristae). The HFR morphology captured in the absence of Dnm1, Mdv1 or Fis1 is thus clearly distinct from the Ringo morphology seen in *WT* cells (also compare Supplementary Movies 1 and 2).

## The Ringo morphology is formed in the absence of contacts between mitochondria and the Endoplasmic Reticulum

Another intriguing distinction between HFR and Ringo networks relates to the presence of ER at constriction sites (Fig. 2i, j and Supplementary Fig. 6f). Constriction sites of hyperfused and HFR mitochondria were systematically decorated with ER membranes, which are known to be required to initiate mitochondrial fission[7]. In contrast, the ER was only rarely detected in the vicinity of the numerous constrictions of Ringo networks (Fig. 2i, j and Supplementary Fig. 6f). These results suggest that Ringo constrictions require Dnm1 but no contact with the ER.

In yeast, contact sites between the ER and mitochondria are mediated by the ERMES (Endoplasmic Reticulum and Mitochondria Encounter Structures) complex that includes four core components – Mmm1, Mdm34, Mdm12 and Mdm10 - where Mmm1 resides in the ER membrane[34]. As a consequence, mitochondrial localization of Mmm1 signals contact sites with the ER. To further assess the roles of Dnm1 and the ER, we analysed the localization of Dnm1-GFP and Mmm1-GFP on mitochondria labelled with Tom70-mCherry under fermentative and respiratory conditions (Fig. 3a; left images). Surprisingly, we observed a two-fold increase in the number of Dnm1-GFP puncta upon respiratory compared to fermentative growth (Fig. 3a; top graph). In contrast, no changes in the number of Mmm1-GFP puncta were observed (Fig. 3a, bottom graph). This difference between Dnm1 and Mmm1 was confirmed by western blot analysis, in which the expression of endogenous Dnm1 increased significantly upon respiratory growth (Fig. 3b; top blot) whereas the expression levels of Mmm1-GFP was unchanged (Fig. 3b, bottom blot). Consistent with this, Dnm1 recruitment on mitochondria increased by more than two-fold in respiration as compared to fermentation, whereas Mmm1 was unchanged (Fig. 3c).

The increase in Dnm1 is transcriptionally regulated, since placing *DNM1* expression under the control of the constitutive TEF promoter abolished Dnm1 variations between fermentation and respiration (Supplementary Fig. 7a). We reasoned that the mitochondrial recruitment of Dnm1 without ER contacts generates stable constrictions as those seen in the Ringo network whereas Dnm1 recruitment at mitochondria-ER (mito-ER) contact sites leads to mitochondrial fission. To test this hypothesis, we tracked mitochondria fission over time using SIM (Supplementary Fig. 7b)[35] and assessed co-recruitment of Mmm1 and Dnm1 at mitochondrial fission sites (Fig. 3d, e). This experiment revealed that the rate of mitochondrial fission is identical in fermentation (where tubular mitochondria dominate) and in respiration (where Ringo networks dominate), with on average 0.28 fission events per cell over 3 minutes of acquisition (Supplementary Fig. 7b). Quantification of Mmm1-GFP and Dnm1-mCherry on more than 200 fission sites also confirmed that both proteins co-localize in the vast majority (about 80%) of fission events (Figs. 3d, e). These results confirm that Dnm1 recruitment at mito-ER contact sites correlate with effective mitochondrial fission. In contrast, Dnm1 recruitment at mitochondria but not at mito-ER contacts would generate regular constrictions forming the Ringo networks. We evaluated this further by quantifying the amount of Dnm1 and Mmm1 puncta at mitochondrial constriction and non-constriction sites (Fig. 3f left images). Importantly, we observed that Dnm1 localization at non-constriction sites is significantly lower in respiration than in fermentation, and its localization at the junction of rings is more than double in respiration than in fermentation (Fig. 3f, left graph). In contrast, no such redistribution between fermentation and respiration was observed for Mmm1 (Fig. 3f, right graph). These results support a mechanism in which Dnm1 triggers repetitive constrictions within the Ringo mitochondrial network that do not result in fission in the absence of contacts with the ER.

## Inhibition of Ringo formation correlates with decreased respiration

Respiring cells require increased mitochondrial protein import compared to fermenting cells[36]. We therefore predict that in galactose or raffinose media that allow for the presence of both tubular and Ringo morphologies (Supplementary Fig. 2e), Ringo networks are associated with enhanced import than tubular networks. To test this prediction, we used a preCox4-mCherry construct as a marker for mitochondrial import efficiency[37] and performed dual color imaging with Tom70-GFP upon growth in galactose and raffinose media. We observed that mitochondrial mCherry intensity is 20% to 45% higher in cells with Ringo than cells with tubular networks (Supplementary Fig. 7c), thereby providing yet another strong hint that the Ringo phenotype is associated with respiration.

To further explore the functional relationships between the Ringo morphology and respiration, we tested if blocking oxidative phosphorylation would prevent the Ringo phenotype. Treatment of *WT* cells with 1 μM Antimycin A, an inhibitor of the respiratory chain acting on complex III[38], did not affect tubular morphology in dextrose media (Supplementary Fig. 8a, left graphs) but induced extensive

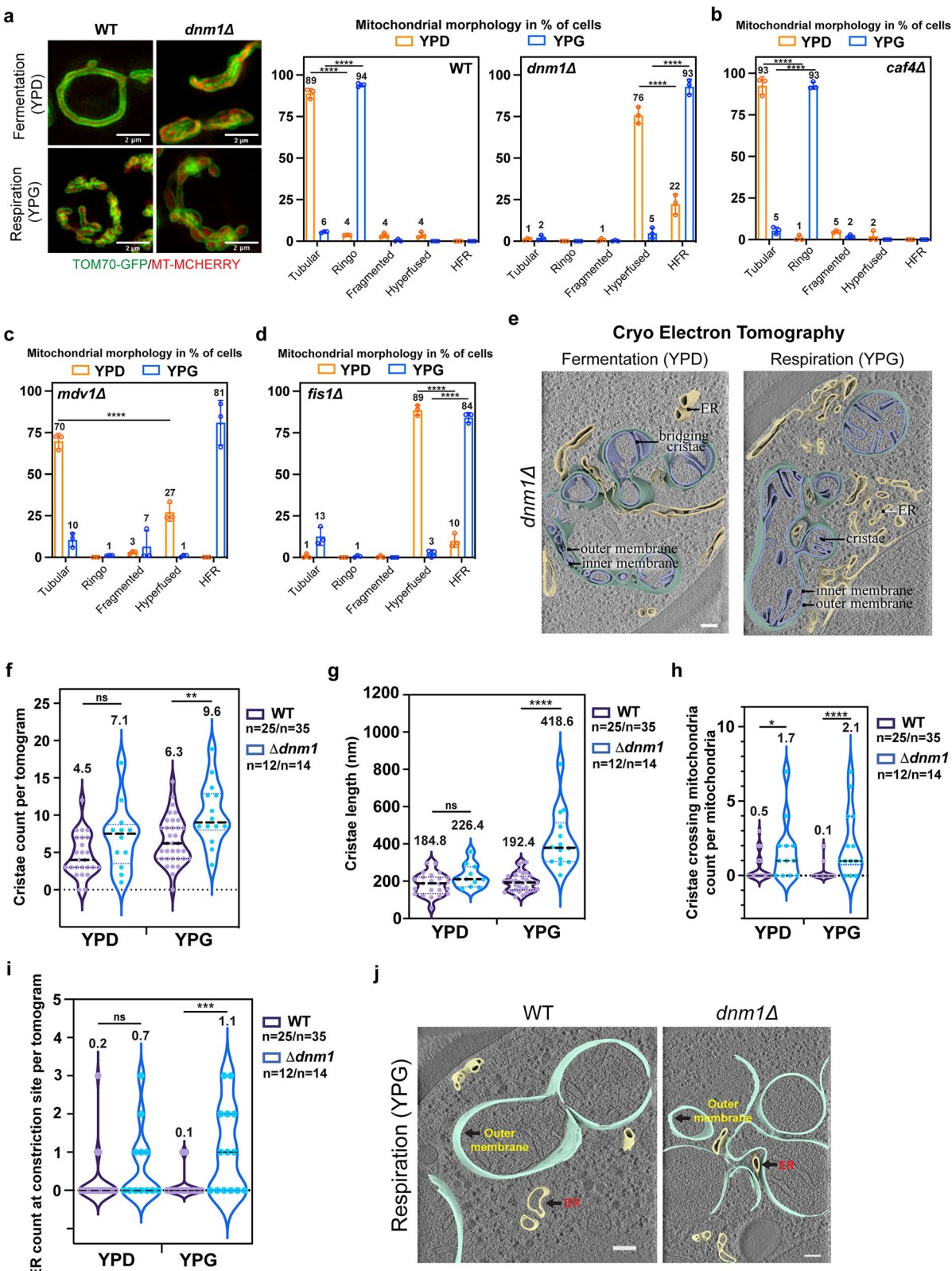

fragmentation of the Ringo network in glycerol already 15 minutes after treatment (Supplementary Fig. 8a, right graphs). Notably, fragmented mitochondria gradually fused into tubular networks after 60 min of treatment but the Ringo morphology was not recovered in this timeframe (Supplementary Fig. 8a, right graph). These results suggest that oxidative phosphorylation is essential for maintaining the Ringo morphology.

Conversely, we asked whether blocking the formation of Ringo networks could correlate with a respiration defect. For this purpose, we relied on our observation that absence of Dnm1 induces disorganization of the Ringo network, resulting into the HFR morphology (Fig. 2). We observed that *dnm1Δ* cells display impaired growth on glycerol media (Fig. 4a; SG). Notably, reintroduction of Dnm1 in *dnm1Δ* cells (Supplementary Fig. 8b) not only restored respiratory growth

**Fig. 2 | Identification and characterization of the HFR mitochondrial morphology. a**–**d** SIM acquisitions (z-projections) of *WT* or *dnm1Δ* cells (a), *caf4Δ* (b), *mdv1Δ* (c) and *fis1Δ* (d) cells labeled for mitochondrial matrix (mt-mCherry) and Outer Membranes (Tom70-GFP) in fermentation (top) or respiration (bottom). Scale bar, 2 μm. Graphs: percentage of cells with Tubular, Ringo, Fragmented, Hyperfused or HFR mitochondria during fermentation (YPD, orange) or respiration (YPG, blue). Mean ± s.d. from > 69 (**a**), > 62 (b), > 78 (**c**) or > 113 (**d**) cells per *n* = 3 independent experiments (colored circles). ***p < 0.0002 (**c**) or ****p < 0.0001 (**d**) (Two-way Anova test followed by Tukey's multiple comparisons test) (**e**) Slices through tomographic volumes and 3D renderings of Hyperfused or HFR mitochondria from *dnm1Δ* cells during fermentation or respiration, respectively. Endoplasmic Reticulum (ER), Nucleus (N) and mitochondrial sub-compartments are indicated. Scale bars, 500 nm. **f** Cristae count per tomogram, *p* = 0.0700 (ns), **p = 0.0066 (**g**) Cristae length, *p* = 0.1233 (ns), ****p < 0.0001 (**h**) Cristae crossing mitochondria count per mitochondria, *p = 0.0241, ****p < 0.0001 and (**i**) ER count at constrictions sites per tomogram (*p* = 0.0694 (ns), ***p = 0.0003) in *WT* or *dnm1Δ* during fermentation (YPD) and respiration (YPG), as quantified in Cryo-ET. In (**f**–**i**), violin plots from *n* = 25 (*WT YPD*), *n* = 35 (*WT YPG*), *n* = 12 (*dnm1Δ YPD*) or *n* = 14 (*dnm1Δ YPG*) tomograms with mean indicated at the top and by dashed lines (quartiles with dotted lines). ns, not significant (Two-tailed Mann-Whitney t-test). **j** Slices through tomographic volumes and 3D renderings of Ringo (*WT*) and HFR (*dnm1Δ*) mitochondria during respiration. Endoplasmic Reticulum (ER), and mitochondrial outer membranes are indicated. Scale bars, 500 nm.

(Fig. 4a; SG) but also reversed mitochondrial morphology from HFR (Supplementary Fig. 8c) to Ringo networks (Fig. 4b). The remaining aggregated, HFR and hyperfused mitochondria are likely caused by random loss of the *DNM1* plasmid from *dnm1Δ* cells (Fig. 4b). These results indicate that inhibition of both respiratory capacity and Ringo morphology formation are exclusively caused by the absence of Dnm1. Yet, they do not allow distinguishing on whether the defective respiratory growth phenotype of *dnm1Δ* cells correlates with abrogation of the Ringo morphology or with the abrogation of mitochondrial fission.

Interestingly, the 8 kDa protein Atg44/Mdi1/Mitofissin, which localizes in the mitochondrial inter-membrane space, was recently shown to participate in completion of mitochondrial fission[39–41]. In *atg44Δ* cells, Dnm1 is recruited to mito-ER contact sites but effective division of mitochondrial tubules is abolished[39–41]. Atg44 thus participates in completion of mitochondrial fission from the inner face of the organelle, which prompted us to investigate its involvement in the formation of Ringo networks. In fermentation and as expected, absence of Atg44 induced accumulation of hyper-fused mitochondrial networks with average mitochondrial diameters equivalent to those from *dnm1Δ* cells (Fig. 4c–e; YPD). In respiratory conditions and more remarkably, lack of Atg44 generated a majority of mitochondrial networks with regular constrictions reminiscent to those seen in *WT* cells and with average diameters closer to those of Ringo than HFR morphologies (Fig. 4c–e; YPG). This Ringo-like morphology that likely correspond to Ringo networks with abrogated fission was largely dominant over HFR networks (Fig. 4e). Consistent with this formation of Ringo-like networks, *atg44Δ* cells were not affected for respiratory growth as opposed to *dnm1Δ*, *mdv1Δ* and *fis1Δ* cells with HFR networks (Fig. 4f and Supplementary Fig. 8d). Taken together, these results indicate that abrogation of mitochondrial fission in *atg44Δ* cells generates Ringo-like mitochondrial network and does not affect respiratory growth. This suggests that the respiratory defect seen in *dnm1Δ*, *mdv1Δ* and *fis1Δ* cells is unlikely correlated to the inhibition of mitochondrial fission but rather to an alternative function of Dnm1 that involves its recruitment to mitochondria.

To confirm this point, we reasoned that decreasing instead of abolishing the expression of Dnm1 could bypass defects in mitochondrial fission and other functions mediated by Dnm1 but may maintain the inhibition of the Ringo morphology formation. We thus placed the *DNM1* gene at its genomic locus under the control of the *ADH1* promoter, which is documented to be activated under fermentation but repressed under respiration[42]. As compared to *WT* cells, *ADH*-driven expression of Dnm1 increased by two-fold in fermentation (SupplementaryFig. 9a) which induced weak formation of fragmented (16%) and Ringo (12%) mitochondrial networks (Fig. 5b, SD, red bars) but did not affect fermentative growth (Supplementary Fig. 9b, SD plates). Upon respiration and as expected, *ADH*-driven expression of Dnm1 was repressed by 50% (Fig. 5a). Since Dnm1 expression increases by two-fold during respiratory growth (Fig. 3a), the two-fold repression induced by the *ADH1* promoter thus brings expression of endogenous Dnm1 at the level seen during fermentation. Consistent with this, *ADH*-driven expression of Dnm1 did

not induce formation of HFR networks, indicating that mitochondrial fission was not affected (Fig. 5b, SG, red bars). However, while the Ringo morphology was no longer dominant and decreased by 60% as compared to *WT* cells, 48% of cells exhibited a tubular morphology (Fig. 5b, SG, red bars). These results indicate that the two-fold increase of Dnm1 during respiratory growth is essential to promote efficient Ringo formation but is not required for mitochondrial fission.

Because of the remaining 32% cells with Ringo networks (Fig. 5b, SG, red bars), serial dilution assays were not sensitive enough to detect any respiratory growth defect of *ADH-DNM1* cells at 23 or 30°C (Supplementary Fig. 9b, SG plates). Nonetheless, cell stress at 37°C revealed that the partial inhibition of Ringo morphology correlates with a decreased capacity of *ADH-DNM1* cells to grow in respiratory conditions (Fig. 5c). We thus resorted to microfluidics approaches to improve the sensitivity in analysing the mortality from *WT* and *ADH-DNM1* cells at 30°C. This led to demonstrate that the 60% inhibition of Ringo formation in *ADH-DNM1* cells at 30°C (Fig. 5b, SG, red bars) correlates with twice more cell death as compared to *WT* cells in respiratory condition (Fig. 5d and Supplementary Fig. 9c). Consistent with this, Oxygen consumption measurements by high-resolution respirometry of *WT*, *dnm1Δ* and *ADH-DNM1* cells grown in Ethanol media at 30°C, revealed that *ADH-DNM1* and *dnm1Δ* cells have an equivalent 40% Oxygen Consumption Rate (OCR) decrease as compared to *WT* cells (Fig. 5e). This decrease in respiration correlates perfectly with the inhibition of Ringo formation as this defect was exclusively observed in respiratory media (Fig. 4g) but not in fermentative media (Supplementary Fig. 9d). During *ADH*-driven expression of Dnm1 in respiratory growth, the dynamin is not tagged and is expressed at levels seen during fermentation, which prevents affecting Dnm1-mediated functions such as mitochondrial fission. Decreased expression of Dnm1 does not affect mitochondrial fission but we cannot rule out it may impact other functions than inhibiting Ringo networks formation. Yet, expression of Dnm1 at levels seen during fermentation affects formation of the Ringo mitochondrial morphology to generate tubular networks which correlates with partial inhibition of respiration.

## Inhibition of Ringo formation correlates with perturbed mitochondrial DNA distribution

To further evaluate the correlation between the Ringo phenotype and respiration, we assessed the expression of distinct subunits of OXPHOS complexes in *WT*, *dnm1Δ* and *ADH-DNM1* cells upon fermentative and respiratory conditions at 30°C. We monitored the expression of Cox1, Cox2 and Cox4, three distinct OXPHOS subunits of complex IV, and Por1, an outer membrane mitochondrial porin (Fig. 6a–d and Supplementary Fig. 10a–c). We observed that while expression of Por1 and Cox4 was unaffected by either *ADH*-driven expression of Dnm1 (Fig. 6a, b) or *DNM1* ablation (Supplementary Fig. 10a), the levels of Cox1 (Fig. 6c and Supplementary Fig. 10b) and Cox2 (Fig. 6d and Supplementary Fig. 10c) were significantly decreased in respiratory conditions. The disruption of complex IV stoichiometry correlates with the inhibition of Ringo formation and may explain the

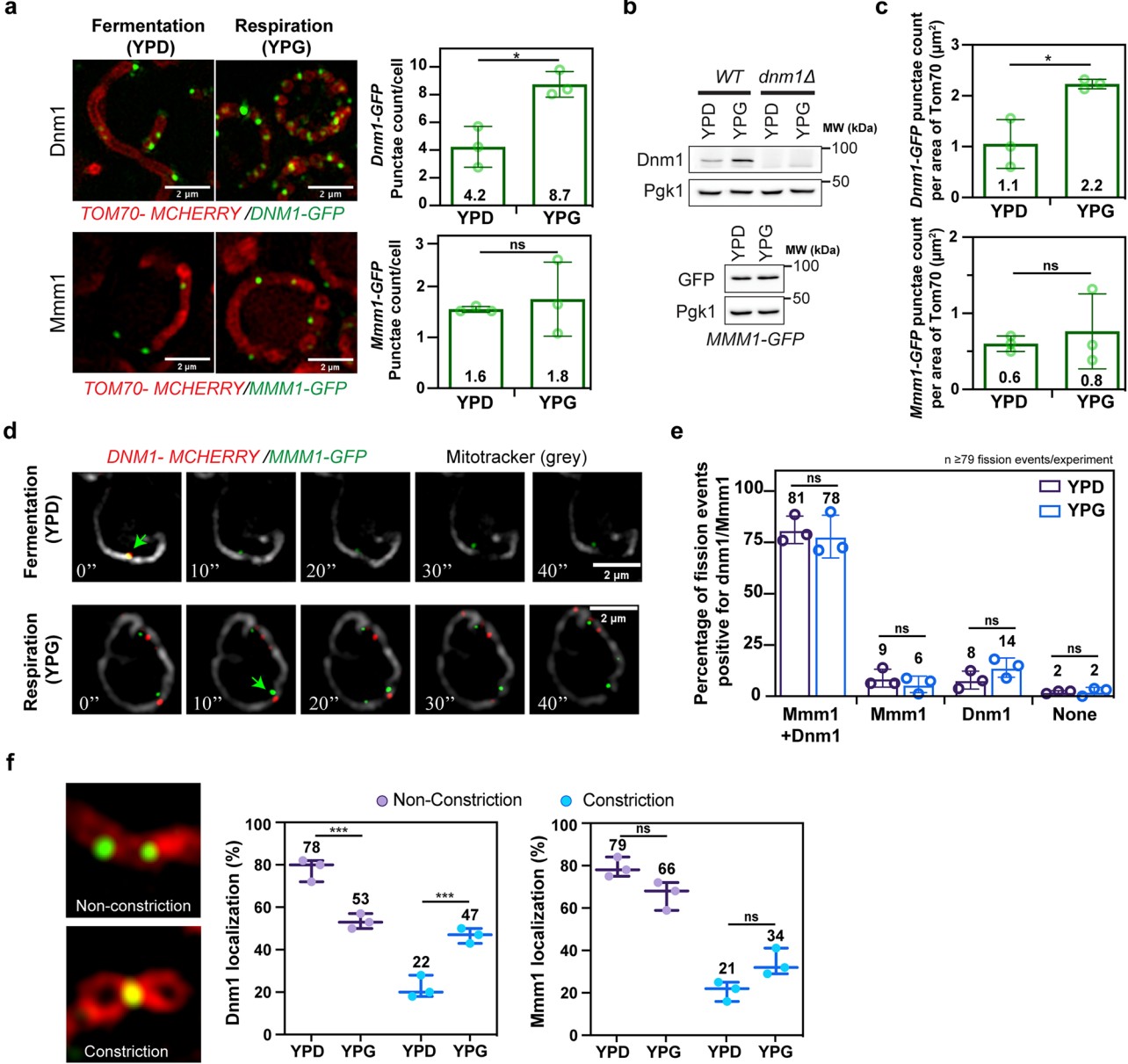

**Fig. 3 | Constrictions within Ringo mitochondria are mediated by Dnm1 but do not require contacts with the ER. a** SIM of *WT* cells labeled for Tom70-mcherry and Dnm1-GFP (top) or Mmm1-GFP (bottom) in fermentation (YPD) or respiration (YPG). Scale bar, 2 μm. Right graphs: Dnm1 (top) and Mmm1 (bottom) punctae count per cell (> 51 for Dnm1; > 77 for Mmm1) in YPD or YPG. *p = 0.0109, ns p = 0.6582 (Two-tailed unpaired t test). **b** Total protein extracts prepared from *WT* and *dnm1Δ* (top) or *MMM1-GFP* (bottom) cells in YPD or YPG and analyzed by immunoblotting as indicated. MW (kDa) indicated on the right. **c** Quantification of Dnm1-GFP (top) or Mmm1-GFP (bottom) puncta count/area of Tom70-mcherry from cells (> 50 cells per experiment) in (a). *p = 0.014, ns p = 0.6056 (Two-tailed unpaired t test).

**d** SIM Time-lapse from Dnm1-mcherry, Mmm1-GFP, mitotracker (grey) triple-labeled cells during fermentation and respiration. Green arrows indicate fission events. **e** Percentage of total fission events positive or negative for both Dnm1 and/ or Mmm1 (> 79 fission events per experiment). ns p = 0.9946, ns p = 0.9964, ns p = 0.8602, ns p > 0.9999 (Two-way Anova followed by Tukey's multiple comparisons test). **f** Percentage of Dnm1 (left) or Mmm1 (right) localization at non-constrictions (purple) or constriction (blue) sites in cells (> 76 cells per experiment) from (a). ***p = 0.0007, ***p = 0.0007, ns p = 0.0930, ns p = 0.0839 (Two-way Anova followed by Tukey's multiple comparisons test). For all graphs, Mean ± s.d. from n = 3 independent experiments (circles and dots). ns, not significant.

requirement of Dnm1 increased expression for optimal respiration (Fig. 5e). Yet, a more intriguing observation is that Cox1 and Cox2 are encoded by mitochondrial DNA (mtDNA) whereas Cox4 and Por1 are encoded by the nuclear genome. Moreover, mtDNA copy number is known to drastically increase in respiratory as compared to fermentative conditions[14]. This suggests that Dnm1 increased expression may regulate mtDNA homeostasis during respiration.

We initially visualized mtDNA using Hoechst-based point accumulation in nanoscale topography (PAINT), a super-resolution technique suited to imaging DNA[43]. This analysis confirmed that mtDNA

significantly increases upon respiration when the Ringo morphology is formed but also that its distribution is strongly affected in the absence of Dnm1 where the Ringo morphology does not form (Supplementary Fig. 10d–f). These results in *WT* and *dnm1Δ* cells were confirmed using a more conventional readout where we tracked Yme2, an integral inner mitochondrial membrane protein that plays a role in maintaining mitochondrial nucleoid structure and number[44] (Supplementary Fig. 10g–i).

To achieve direct visualization of mtDNA in a non-invasive fashion, we then employed the mtLacO-LacI system, which allows to image

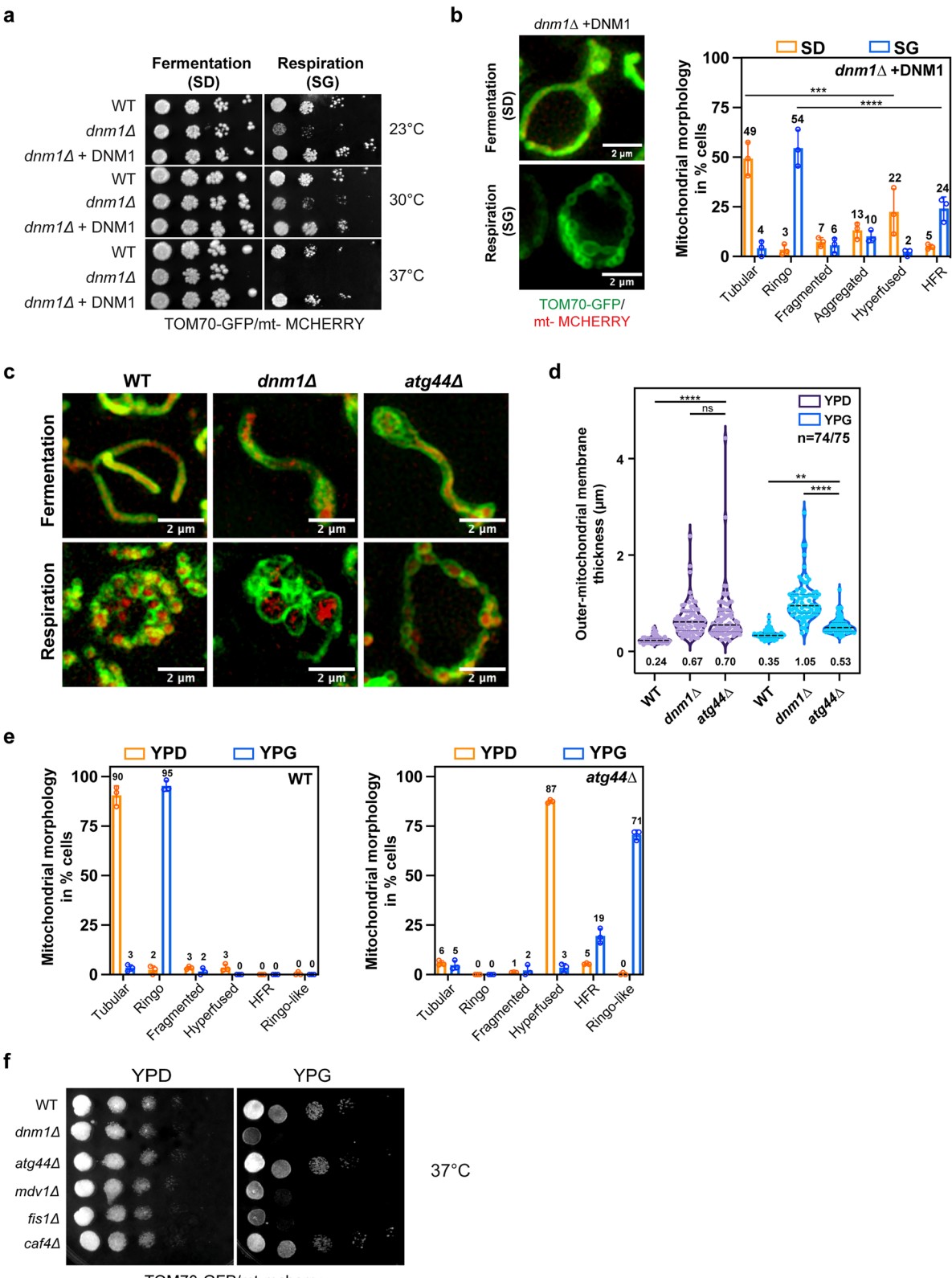

mtDNA dynamics in living cells[45]. In this system, mtDNA has been modified with a synthetic array consisting of 11 LacO repeats introduced upstream of the COX2 gene[45]. In parallel, a triple GFP-tagged mitochondrially targeted LacI protein (mt-3xGFP-LacI) can bind to LacO repeats leading to clearly discernible foci that distribute throughout the mitochondrial network[45]. We employed this system under fermentative and respiratory conditions in *WT* and *dnm1Δ* cells

but most importantly under the *ADH-DNM1* context to evaluate the impact of Dnm1 decreased expression on mtDNA abundance and distribution. Staining of mtLacO-LacI strains with mito-tracker allowed efficient visualization of mt nucleoids within the mitochondrial network (Fig. 6e). As expected from the increase in mtDNA copy number upon respiratory as compared to fermentative growth[14], mtDNA localizations significantly increased upon respiration in *WT* cells

**Fig. 4 | Atg44 is dispensable for the Ringo morphology formation. a** Dextrose and glycerol serial dilutions of *WT*, *dnm1Δ* and *dnm1Δ+DNM1* strains at 23, 30 and 37 °C. **b** SIM acquisitions (z-projections) of *dnm1Δ+DNM1* cells labeled with mt-mCherry and Tom70-GFP in fermentation (top) or respiration (bottom). Scale bar, 2 μm. Right graph: percentage of cells with Tubular, Ringo, Fragmented, Aggregated, Hyperfused or HFR mitochondria during fermentation (SD, orange) or respiration (SG, blue). Mean ± s.d. from >64 cells in n = 3 independent experiments (colored circles). ***p = 0.0004, ****p < 0.0001 (Two-way Anova test followed by Tukey's multiple comparisons test) (**c**) Same as (b) with *WT*, *dnm1Δ* or *atg44Δ* cells in YPD (top) or YPG (bottom) media. **d** Average mitochondrial thickness per cell from (**c**), in YPD or YPG as quantified in SIM. Violin plots from n = 74 to 75 cells with mean indicated at the bottom and by black dashed lines (quartiles with dotted lines). ****p < 0,0001,ns p = 0.9966, **p = 0.0089, ****p < 0.0001 (Two-way Anova test followed by Tukey's multiple comparisons test) (**e**) Percentage of *WT* and *atg44Δ* cells from (**c**) with Tubular, Ringo, Fragmented, Aggregated, Hyperfused, HFR or Ringo-like mitochondria during fermentation (YPD, orange) or respiration (YPG, blue). Mean ± s.d. from > 73 cells in *n* = 3 independent experiments (colored circles). **f** Same as (**a**) with *WT*, *dnm1Δ*, *atg44Δ*, *mdv1Δ*, *fis1Δ* and *caf4Δ* cells at 37 °C (see also Supplementary Fig. 8d). For all graphs, ns, not significant.

(Fig. 6f; *WT* green and red bars). This mtDNA increase is unlikely caused by additional mito-ER contacts and mitochondrial fission events[10,11] since both occur at equivalent frequencies in fermentation and respiration conditions (Fig. 3c and Supplementary Fig. 6b). Consequently, if the Ringo morphology correlates with the increase of mtDNA during respiration, this increase should be affected by *ADH*-driven or abrogated expression of Dnm1. The increase in mtDNA loci seen in *WT* cells during respiration with Ringo networks (Fig. 6f; *WT* orange and green bars) was decreased upon partial (*ADH-DNM1*) and totally abolished upon total (*dnm1Δ*) inhibition of the Ringo morphology formation (Fig. 6f; green bars). Similarly, the distribution of mt nucleoids along the mitochondrial network upon respiratory growth (Fig. 6g; green bars) was significantly impacted in *ADH-DNM1* (1.3 GFP puncta/μm² of mitotracker) and *dnm1Δ* cells (0.8 GFP puncta/ μm² of mitotracker) as compared to *WT* cells (1.9 GFP puncta/μm² of mitotracker). These data with the mtLacO-LacI system (Fig. 6e–g) confirm those obtained with the Hoechst-PAINT (Supplementary Fig. 10d–f) and Yme2-GFP systems (Supplementary Fig. 10g–i). Importantly, they also validate that the effects seen in *dnm1Δ* cells can also be observed in *ADH-DNM1* cells, comforting a close link between Dnm1 increased expression and regulation of the amount and distribution of mtDNA during respiration. Taken together these results show that the inhibition of Ringo formation correlates with perturbation of mtDNA distribution during respiratory growth.

## Discussion

The relationship between mitochondrial morphology and respiration in fungi is the subject of an ongoing debate[12,13,19,20]. However, the limitations of confocal microscopy in characterizing mitochondrial morphology at high-resolution have thus far hindered a clear understanding of the impact of mitochondrial morphology on oxidative phosphorylation[12,13]. Our results reveal that regular constrictions mediated by Dnm1 along the mitochondrial network during respiration give rise to the formation of a previously unknown morphology which we refer to as "Ringo". As measured by SIM, the average length of Ringo networks is 29.6 ± 10 μm *vs* 15 ± 5 μm for tubular networks (Supplementary Fig. 5b). Ringo mitochondria are also larger with average mitochondrial diameters of 0.4 ± 0.1 μm *vs* 0.23 ± 0.02 μm for tubular mitochondria (Figs. 1f, 4d and Supplementary Fig. 6b). Ringo networks are characterized by significantly increased mitochondrial constrictions of the outer membrane with 0.82 ± 0.3 constrictions / μm length of mitochondria *vs* 0.23 ± 0.001 constrictions / μm for tubular networks (Supplementary Fig. 4b). The distance between constrictions of outer membranes in Ringo networks is thus ~1.2 μm.

Previous studies in mammalian cells have reported transient constrictions of the mitochondrial inner membrane but not of outer membranes[46–48]. Constriction of Mitochondrial Inner Compartments (CoMIC) occurs at ER−mitochondrial contact sites and is initiated independently of Drp1 action and mitochondrial outer membrane constriction[48]. CoMIC is induced and potentiated by intra-mitochondrial Ca²⁺ entry and not Drp1 activity[46,48]. This mitochondrial calcium uptake has been shown to be stimulated by INF2-mediated actin polymerization[46]. The distance between constrictions of inner membranes is ~2 μm and 90.4 ± 1.9% of the constrictions correspond to sites of close ER association[46]. Drp1 is not required for the initiation of CoMIC, but likely terminates CoMIC cycles by mitochondrial division[48]. More recently, Hexokinase 1 (HK1), a key glycolytic enzyme, was shown to form rings around mitochondrial outer membranes during energy stress[49]. The formation of HK1-rings is promoted by lack of ATP and glucose-6-phosphate (G6P). HK1-rings do not depend on DRP1 but on the ER and inhibit mitochondrial fission[49]. CoMIC and HK1-rings thus differ from the Ringo morphology in several regards.

Ringo constrictions are mediated by Dnm1 with 8.7 ± 0.7 Dnm1 puncta per Ringo network vs 4.2 ± 1.5 Dnm1 puncta per tubular network and doubled Dnm1 density of 2.2 ± 0.1 puncta/ μm² of Ringo mitochondria vs 1.1 ± 0.5 puncta/ μm² of tubular mitochondria (Fig. 3a, c).Lack of Dnm1 abolishes formation of the Ringo morphology to generate HFR networks (Fig. 2) while decreased expression of Dnm1 during respiratory growth at level seen during fermentative growth inhibits Ringo formation to generate tubular networks (Fig. 5). CoMIC and HK1-rings do not depend on Drp1[46,48,49].

Cryo-ET analysis (Fig. 2i) revealed that the ER is absent from Ringo constrictions (0.1 ER per constriction) which contrasts with abortive fission in HFR networks where each constriction was decorated by ER membranes (1.1 ER per constriction). Consistent with this, mito-ER contacts visualized by Mmm1 localization at mitochondria did not vary between tubular and Ringo networks with 1.8 ± 0.5 Mmm1 puncta per Ringo network *vs* 1.6 ± 0.1 Mmm1 puncta per tubular network and unchanged Mmm1 density of 0.8 ± 0.5 puncta/ μm² of Ringo mitochondria vs 0.6 ± 0.1 puncta/ μm² of tubular mitochondria (Fig. 3a, c). Unlike the Ringo morphology, CoMIC and HK1-rings depend on the ER[46,48,49].

Ringo constrictions do not lead to mitochondrial separation. Yet, mitochondrial fission over 180 sec is equivalent in tubular and Ringo networks with 0.279 ± 0.9 event per Ringo network and 0.282 ± 0.5 event per tubular network (Supplementary Fig. 7b), indicating that mitochondrial fission is not inhibited during Ringo formation. Conversely, abrogation of mitochondrial fission through inactivation of Atg44 does not affect Ringo formation (Fig. 4). In contrast, HK1-rings inhibit mitochondrial fission[49] while fission inhibition through Drp1-KD causes an increase in CoMIC[46].

In terms of function, HK1-rings form during energy stress promoted by lack of ATP and G6P[49], CoMIC cycles have been proposed to take place prior Drp1-mediated mitochondrial division[48] while the formation of Ringo networks correlates with increased respiration. The Oxygen consumption rate from 2.10⁶ *WT* cells with tubular networks during fermentative growth reach 0.373 ± 0.1 pmol/min (Supplementary Fig. 9d). This rate does significantly change upon ablation of *DNM1* with 0.310 ± 0.1 pmol/min for Hyperfused networks during fermentative growth (Supplementary Fig. 9d) or 0.299 ± 0.08 pmol/min for HFR networks during respiratory growth (Fig. 5e). In contrast, *WT* cells with Ringo networks during respiratory growth reach the highest rate of Oxygen consumption with 0.502 ± 0.08 pmol/min (Fig. 5e), while *ADH-DNM1* cells with tubular networks resulting from partial inhibition of Ringo formation remain at 0.287 ± 0.1 pmol/min (Fig. 5e), similar to cells with tubular, Hyperfused or HFR networks.

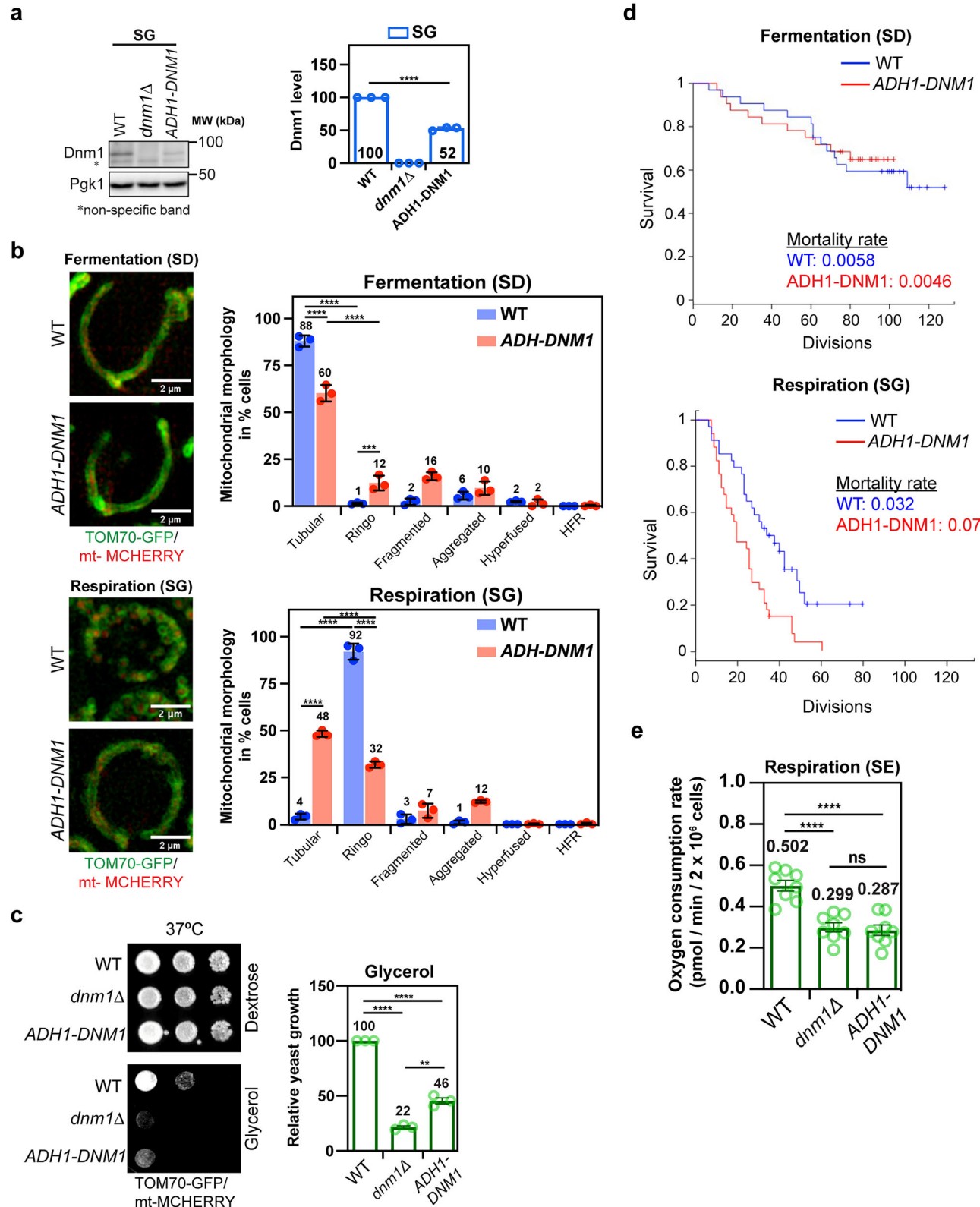

*ADH*-driven expression of Dnm1, led to show that expression of Dnm1 during respiratory growth at levels seen during fermentative growth induces partial inhibition of respiration (Fig. 5e), decreased expression of the mt-DNA encoded subunits of complex IV Cox1 and Cox2 (Fig. 6c-d) as well as perturbed distribution of mt nucleoids (Fig. 6g). In parallel, *ADH*-driven expression of Dnm1 affects formation of the Ringo mitochondrial morphology to generate tubular networks

(Fig. 5b). The possibility that decreased expression of Dnm1 may impact other functions than inhibiting Ringo networks formation is not excluded. However, expression of endogenous and fully functional Dnm1 from the *ADH1* promoter during respiratory growth did not affect mitochondrial fission (Fig. 5b).

Based on considerations above, future investigation will need to prove that the Ringo morphology is causal in promoting respiration by

**Fig. 5 | Inhibition of Ringo formation correlates with decreased respiration.**
**a** Total protein extracts prepared from *WT*, *dnm1Δ* or *ADH1-DNM1* cells in glycerol (SG) media and analyzed by immunoblotting as indicated. MW (kDa) indicated on the right. Right graph: Dnm1 levels normalized to Pgk1 in *dnm1Δ* and *ADH1-DNM1* relative to the *WT* strains in SG. Mean ± s.d. from *n* = 3 independent experiments (blue circles). ****$p$ < 0.0001 (One-way Anova followed by Tukey's multiple comparisons test). **b** SIM acquisitions (z-projections) of *WT* and *ADH1-DNM1* cells labeled with mt-mCherry and Tom70-GFP in fermentation (top) or respiration (bottom). Scale bar, 2 μm. Right graphs: percentage of *WT* (blue) and *ADH1-DNM1* (red) cells with Tubular, Ringo, Fragmented, Aggregated, Hyperfused or HFR mitochondria during fermentation (Top) or respiration (Bottom). Mean ± s.d. from > 95 cells in *n* = 3 independent experiments (colored circles). ****$p$ < 0,0001,

****$p$ = 0.0007 (Two-way ANOVA multiple comparisons) (**c**) Dextrose and glycerol serial dilutions of *WT*, *dnm1Δ* or *ADH1-DNM1* strains at 37 °C. Right graph: Quantification of indicated cells grown at 37 °C on glycerol media relative to the *WT* strain. Mean ± s.d. from *n* = 3 independent experiments. ****$p$ < 0.0001, **$p$ = 0.0014 (Two-tailed unpaired t-test) (**d**) Survival curves derived from single-cell analysis of *WT* (blue) and *ADH-DNM1* (red) cells in SD (top) or SG (bottom) media (Supplementary Fig. 9c), were generated using the Kaplan–Meier estimator (*WT* *n* = 32 cells; *ADH1-DNM1* *n* = 32 cells for SD; *WT* n = 34 cells; *ADH1-DNM1* *n* = 34 cells for SG). **e** Oxygen Consumption Rates (OCRs) of *WT*, *dnm1Δ* and *ADH-DNM1* cells grown in 2% Ethanol media (SE) at 30 °C. Mean ± s.d. from *n* = 8 independent experiments (green circles). ****$p$ < 0.0001, ns $p$ = 0,9299 (Two-tailed unpaired t-test). ns, not significant.

maintaining optimal levels and distribution of mtDNA to favor stoichiometric assembly of OXPHOS complexes. The involvement of mitochondrial constrictions in the regulation of mtDNA amount and distribution is not unknown as mitochondrial fission has been implicated in the propagation of mt nucleoids[10,11]. In this context, Ringo natural constrictions independent of ER membranes may regulate mtDNA distribution during respiration, which would be in line with a Ringo requirement in synthesis of OXPHOS subunits that are specifically encoded by mtDNA (Fig. 6a–d). Consistent with this, it has been shown that each mtDNA copy has a "sphere of influence" and remains spatially linked to its gene products[50]. The Ringo morphology may thus contribute to respiration by promoting accurate distribution of mt nucleoids along the mitochondrial network thereby favouring optimal distribution of *OXPHOS* complexes during respiratory growth. While our data provide hints toward this model, further investigation will be required to confirm a causal relationship between formation of Ringo networks and respiration.

During respiratory growth, constrictions of outer membranes within Ringo networks are accompanied with increased cristae density[16,17,32]. Nonetheless regulation of cristae density and formation of mitochondrial constrictions are promoted by distinct mechanisms as prevention of the Ringo morphology does not block the increase in cristae density (Fig. 2f). The mechanism of mitochondrial constrictions is also distinct from the mechanism of mitochondrial fission, since recruitment of Dnm1 to outer membranes occurs independent of contacts with the ER. The signals that trigger Dnm1 targeting to mitochondria in the absence of ER contacts remain elusive. Yet, the Mdv1 adaptor and mitochondrial anchor Fis1 together with increased expression of Dnm1 during respiration play a role in this process. Intriguingly, a similar increase of the mammalian dynamin DRP1 accompanied with extensive remodelling of mitochondrial morphology has been observed in skeletal muscle during exercise[51]. This suggests that mechanistic and functional features of the Ringo mitochondrial morphology during increased oxidative phosphorylation may be evolutionarily conserved across species.

## Methods

### Yeast culture and transformation
The *saccharomyces cerevisiae* strains used in this study are listed in Supplementary Table 1. Yeast cells were grown and transformed using standard method[52]. In the indicated strains, *DNM1*, *MMM1*, *OM45*, *TOM70* and other genes were chromosomally deleted, C-terminally tagged or had their promoter changed using conventional homologous recombination approaches[53,54]. Where indicated, strains were transformed with plasmids listed in Supplementary Table 2 under selection of interest. Precultures were launched overnight at 30 °C in YP complete or Synthetic minimal media completed with 2% Dextrose. According to the experimental conditions, growth media was switched with distinct carbon sources at 2% Dextrose, Glycerol, Ethanol, Galactose or Raffinose. Cells were diluted to achieve at least 3 cell divisions and reach the OD600 = 0.6 to 0.8 per mL of culture at the time of processing for further experiments.

For culture media switch experiments from Supplementary Fig. 2c, d, cells pre-cultured in YPD/YPG media were diluted into YP media containing distinct dextrose concentrations (0.2%, 0.5%, and 2%) at OD600 = 0.6 per mL of culture. Cells were then allowed to grow before image acquisition at various time points.

For respiration inhibition experiments from Supplementary Fig. 8a, cells were cultured in YPD/YPG media at 30 °C to mid-log phase (OD600 = 0.6-0.7 per mL of culture). AntimycinA (Sigma, A8674) reconstituted in ethanol was then added to the culture at 1 μM concentration. For control samples, equal volumes of ethanol were added to cultures.

We noticed that Synthetic minimal media is essential to promote optimal repression of the *ADH1* promoter under respiratory growth. In consequence, all experiments with the *ADH1-DNM1* strains (Figs. 5;6; and Supplementary Fig. 9; Supplementary Fig. 10) were performed with cells grown in Synthetic minimal media completed with all amino acids and 2% Dextrose, 2% Glycerol or 2% Ethanol.

### Western blot preparation and quantification
Cells were collected during the exponential growth phase and total protein extracts were prepared by the NaOH and trichloroacetic acid (TCA) lysis method[55]. Proteins were separated by SDS-PAGE and transferred onto Membrane Amersham protran 0,45 μm (Amersham, 10600002). The primary antibodies used for immunoblotting were monoclonal anti-Pgk1 (1/20,000, AbCam, ab113687), monoclonal anti-GFP (1/1000, Roche, 11814460001), polyclonal anti-Dnm1 (1/1000, generated by Covalab), monoclonal anti-Por1 (1/10,000, Invitrogen, 459500), anti-Cox1 (1/1000, AbCam, ab110270), anti-Cox2 (1/1000, AbCam, ab110271) and anti-Cox4 (1/1000, AbCam, ab110272). The primary antibodies were detected by incubation with horseradish peroxidase (HRP)-conjugated anti-mouse or anti-rabbit secondary antibodies (Sigma-Aldrich, A9044 and A5278), followed by incubation with the Clarity Western ECL Substrate (Bio-Rad, 1705060). Immunoblotting images were acquired with a Gel Doc XR+ (Bio-Rad) and quantified with the Image Lab 6.0 software (Bio-Rad). The cytosolic protein Pgk1 was used as a loading control to normalize loading of other proteins relative to the WT conditions. Uncropped blots can be found in the Supplementary Information.

### Image and statistical analysis
For image analysis, we used ImageJ[56] and wrote dedicated macro to do the analysis. This Macro was used for all the sample in the same experiment. Results are shown as mean ± standard error of the mean (s.e.m.) for experiments with more than 8 datapoints or Standard Deviation (s.d.) for experiments with less than 8 datapoints[57]. Graphs were created and statistical analysis was performed using Graphpad Prism (v8.0) or Microsoft Excel. The specific statistical tests are indicated in the figure legends. *P* value corresponds to *$P$ < 0.05, **$P$ < 0.01, ***$P$ < 0.001, ****$P$ < 0.0001, ns, not significant.

### Sample preparation for live cell imaging
Yeast cells were harvested upon reaching the exponential phase (OD600 = 0.6–0.8 per mL of culture). 1 mL of cell culture was

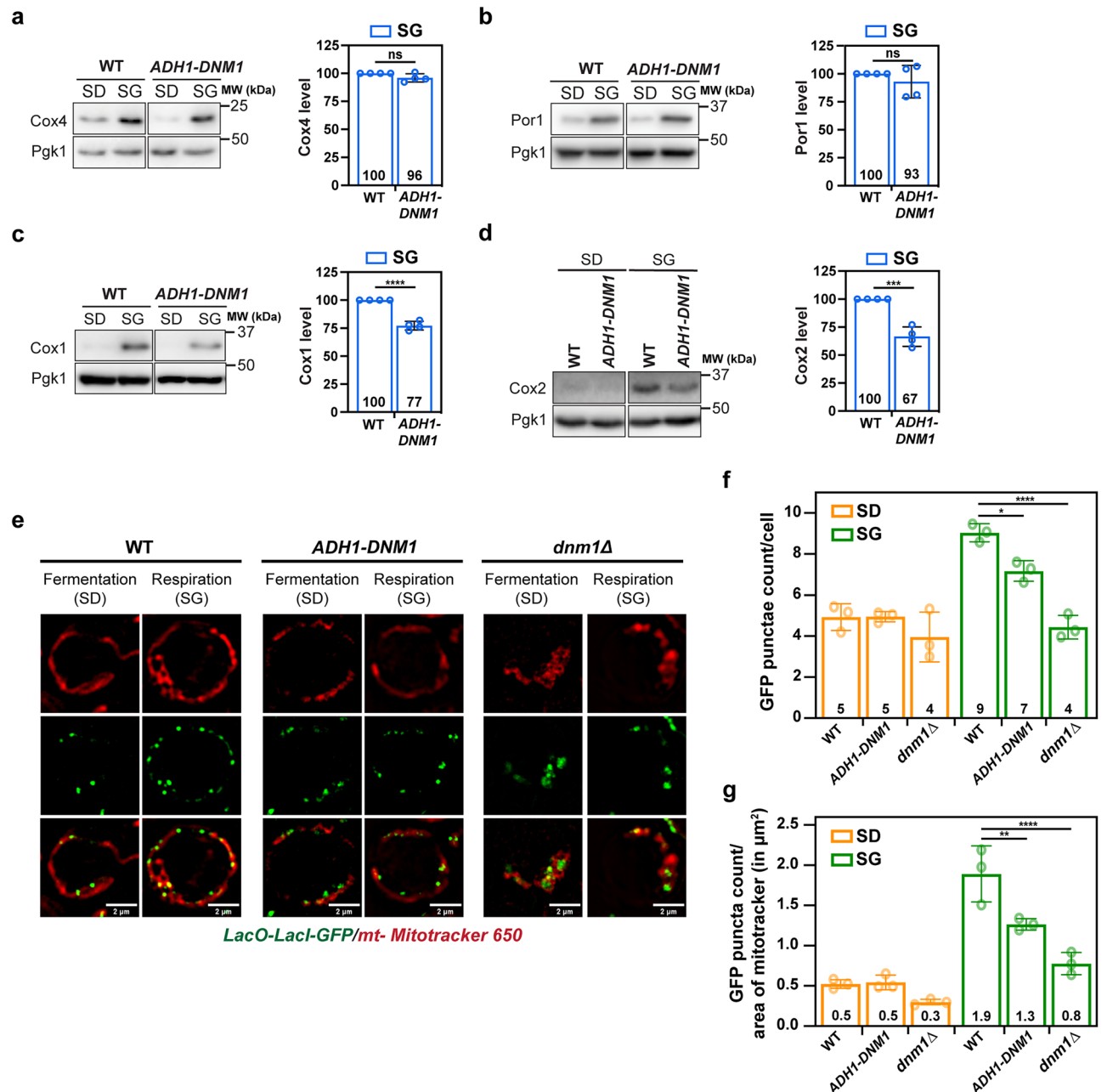

**Fig. 6 | Inhibition of Ringo formation correlates with decreased OXPHOS components expression and perturbed mtDNA homeostasis. a–d** Total protein extracts prepared from *WT* and *ADH1-DNM1* cells in SD or SG media analyzed by immunoblotting as indicated. MW (kDa) on the right. Right graphs: Cox4 (**a**), Por1 (**b**), Cox1 (**c**) and Cox2 (**d**) levels normalized to Pgk1 in *ADH1-DNM1* relative to the *WT* strains in SG media. Mean ± s.d. from *n* = 4 independent experiments (blue circles). ns *p* = 0,0688 (a), ns *p* = 0,3672 (b), ****p* < 0,0001 (c) and ***p* = 0,0002 (d), ns, not significant, (Two-tailed unpaired t-test). (e) SIM acquisitions of *WT*, *ADH1-*

*DNM1* and *dnm1Δ* mtLacO-LacI strains (LacI-GFP) in SD or SG media and labelled with Mitotracker 650. Scale bar, 2 μm. **f** LacI-GFP puncta count per cell in *WT*, *ADH1-DNM1*, and *dnm1Δ* strains ( > 22 cells) during fermentation (SD, orange) and respiration (SG, green). ***p* = 0,0484, *****p* < 0,0001 (**g**) Quantification of LacI-GFP puncta count/area of Mitotracker 650 from cells ( > 22) in (**e**). **p* = 0.005, *****p* < 0,0001. In (**f**) and (**g**), Mean ± s.d. from *n* = 3 independent experiments (colored circles), two-way Anova followed by Tukey's multiple comparisons test.

centrifuged and resuspended in 30 μL of fresh media. 8 μL of cell suspension was gently placed on 25 mm glass coverslip (#CG15XH, Thorlabs, Germany). Cells were immobilized using a small agar pad containing culture media (*e.g.*, YPD, YPG, SD, SG etc. depending on the experimental condition) which was placed gently on the cell suspension. This resulted in a monolayer of immobilized yeast cells optimal for live cell imaging. The coverslip was ultimately placed in the 25 mm magnetic chamber (#CM-B25-1, Chamlide magnetic chambers) for imaging.

**Spinning disc microscopy**

Spinning disk cross-sectional images (Fig. 1a) were acquired using Spinning-disk UltraView VOX (Perkin Elmer) system equipped with a confocal scanning head (CSU X1; Yokogawa), a 100 × 1.4 NA oil immersion objective, and EMCCD cameras (ImageEM C9100; Hamamatsu Photonics) controlled by Volocity software. The GFP and mcherry channel images were acquired using 488 and 561 lasers with 10% and 25% of laser power, respectively, and a maximum of 300 ms of exposure time. The time-lapse movies (Supplementary Fig. 7b;

Fig. 3d, e) were recorded for 3 minutes at 10-second intervals. Post-processing of images was performed using ImageJ (National Institute of Health) open-source software.

## Structured illumination microscopy

Cells were imaged using commercial Zeiss LSM 780 Elyra microscope (Carl Zeiss, Germany) controlled by the Zen software. The microscope was equipped with an oil immersion 100x Plan-Apochromat objective with a 1.46 numerical aperture and an additional 1.6x lens. For detection, an EMCCD Andor Ixon 887 1K camera was used. One SIM image was reconstructed from nine images acquired from three different phases and three different angles. Acquisition parameters were adapted to optimize the signal to noise ratio according to yeast strains and protein under investigation. For z-stack acquisition, we kept the z-interval between 80-150 nm depending on the strain and experimental requirement. SIM images were reconstructed with ZEN software and then channel alignment was performed using 100 nm TetraSpeck fluorescent beads (Cat#T7279, Invitrogen) embedded in the same conditions as the sample.

## 3D reconstruction of SIM stacks

3D rendering of SIM stack of Tom70-GFP strains (Supplementary Movies 1 and 2) was preformed using ImageJ plugins, 3D Objects Counter and 3D suite[58]. We wrote a macro to apply the same operation on all the stacks. The macro performs the following operation to subtract the background: two duplicate images were created and a 3D-Gaussian blur of 1 and 3 pixels was applied and then images were subtracted. The Otsu threshold of the background subtracted stack was measured. An object image was generated using a 3D object counter plugin with Otsu thresholding and a minimum size filter of 25. The object image was imported to a 3D suite plugin using the 3D manager and visualized using the 3D viewer plugin. Further, the surface was smoothed by 7 iterations. Finally, a 360-degree rotation movie was recorded.

## 3D skeleton analysis of SIM stacks

Mitochondrial networks skeleton analysis in Supplementary Fig. 5 was performed as follows. We wrote a customized macro in ImageJ to apply the same operation on all data sets. The raw SIM stack of Tom70-GFP strain was smoothened once and then a 3D-Gaussian blur filter of 3 pixels was applied. Using Otsu thresholding, the stack was converted into a binary stack. This binary stack is duplicated and projected to check the presence of any signal from a neighboring cell. If any unwanted signal is present then it is manually removed using a clear filter. Using the "Skeletonize (2D/3D)" plugin, the stack is skeletonized. Finally, measurement was performed using the "Analyze skeleton" plugin with the Prune (none) setting. Result table was imported to MS Office Excel and relevant data were extracted and plotted on GraphPad Prism software.

## Mitochondrial morphology quantification

In Figs. 1a, b, 2a–d, 4b, 4e, 5b, Supplementary Fig. 1c, 2, 8a and 8c, we quantified the morphology visually. A z-stack of images was converted into z-projection image using Maximum intensity projection. Each field contained two channels, one for OM (Tom70-GFP) and one for matrix (mt-mcherry). We mostly focused on GFP labelling of OM. Tubular morphology was identified by two parallel threads forming oval shaped close to cortex of cell. Ringo morphology was identified by series of rings connected throughout the cortex of cells, which appear like Tom70 localized in spiral fashion around the matrix. Ringo morphology was far denser and had branched networks (Supplementary Fig. 5). Fragmented mitochondria were easily identified by separated rings, localized mostly at the center of a yeast cell. Hyperfused mitochondria appeared much thicker than tubular, and partially covered the cells unlike tubular or Ringo mitochondria. It appeared significantly straight structure and localized at the edges of cell. Finally,

HFR morphology was easily identified by few uneven size rings, much thicker than Ringo mitochondria. HFR morphology partially covered the cells like Hyperfused mitochondria. From each field we calculated cells with different morphology and compared as percentage of total in a population of cells.

## Photo activated localization microscopy (PALM)

For PALM imaging, (Fig. 1c and Supplementary Fig. 1b), we used an inverted Nikon Ti-E Eclipse microscope equipped with a 100 × 1.49 NA objective (Nikon) with the Perfect Focus System active. A 488-nm wavelength laser with 200 mW power (at 1%) was used to acquire low resolution images before activation using blue light source. Blue LED light source emitting at 405 nm wavelength was used to the photo activation of mEOS. A 561-nm wavelength laser with 500 mW (at 75%) was used to acquire 50,000 images with an exposure time of 50 ms. Images were acquired on an EMCCD camera with 1.5x additional magnification, resulting in pixel size of 106 nm and a Field of View (FoV) of 54 μm × 54 μm. EM gain of the EMCCD was set to 300. 3.7% formaldehyde (#8.18708, Sigma-Aldrich) fixed yeast cells in which Tom70 is tagged with mEOS2 was mounted on the coverslip using collagen gel. Briefly, 3.6 mg/ml collagen stock (#354236, Corning) was diluted to final matrix concentrations of 2.2 mg/ml and neutralized with 1 M sodium hydroxide (#28-3010, Sigma-Aldrich). Cells were mixed into this collagen solution and a drop was mounted on the top of the coverslips and allowed to polymerize at RT for 30 minutes. Collagen gel was used to immobilize yeast cells because agar pad introduced significant background and non-specific blinking events.

## Single molecule localization microscopy (SMLM) image reconstruction

For SMLM data reconstruction (PALM and Hoechst PAINT), the ThunderSTORM[59] plugin available in ImageJ[56] was used. We chose Wavelet filter (B-Spline) for image filtering, local maximum method for approximate localization of molecule. PSF: Gaussian method was used for sub-pixel localization of molecules with weighted least squares fitting method. We either used fluorescent beads or cross-correlation to perform the drift correction. We filtered out localization that have sigma values below 100 or above 200, or localization that have uncertainty above 50 nm. Localization appearing in consecutive frames separated by less than 60 nm was merged, with the assumption that they are originating from the same molecule. Finally, localization table was saved into ".CSV" format and Normalized Gaussian method was used for image visualization. Final image resolution was in the range of 40–50 nm.

## Transmission Electron microscopy

Yeast cells were processed from adapted OTO protocol[60]. Briefly, cells were grown in YPD/YPG media and were fixed by immersion in 2.5% glutaraldehyde, and post-fixed. First in a mix of 1% OsO4 with 1.5% K4Fe(CN)6 (potassium ferrocyanide–reduced osmium) to increase the membranes contrast, then contrast was enhanced by tannic acid 1%. A second post-fixation was performed in 1% OsO4. Subsequently, the samples were dehydrated through increasing graded ethanol series and embedded in epoxy resin (Electron Microscopy Sciences) at 60 °C for 48 hrs. The embedded resin block was mounted on a specimen stub using colloidal silver paint (Electron Microscopy Sciences) and sputter coated with 200 nm of gold/palladium. Embedding samples were cut with an ultramicrotome. Ultrathin sections (70 nm) were cut with a Leica Ultracut S microtome and stained with uranyl acetate followed by lead citrate. Contrasted sections were observed using a TEM FEI Tecnai T12 at 120 kV.

## TEM images quantification

To generate the graphs in Figs. 1f, g from TEM images, we quantified mitochondrial diameter by drawing a line randomly from one end to

the other end of mitochondria. The length of the line was then measured. For final plotting, an average of 3 random lines per cell was considered. To count the number of rings and constriction, we visually counted the structure/sites on mitochondria such as those denoted by green and yellow arrows in Fig. 1d and Supplementary Fig. 4a. Circularity of mitochondria was set as an important factor for ring characterization. We thus considered not counting oval shaped mitochondria as rings. For constrictions, continuity in the mitochondrial network was the main factor along with the small diameter of mitochondrial tubules.

## Cryo-electron tomography

For grids preparation, *S. cerevisiae* cells (*WT* or *dnm1Δ*) were grown in either YPD or YPG media to an OD600 = 0.9–1.3 μl of cell suspension was applied to a freshly grown discharged holey carbon grid (Quantifoil® #200, R2/2). Grids were immediately blotted from the back side using Whatman filter paper #1 (GE Healthcare) within the automatic plunge freezer EM GP (Leica Microsystems). Grids were stored at liquid nitrogen temperature until further use.

Lamellae were prepared using an Aquilos FIB-SEM system (Thermo Fisher Scientific). Grids were coated with an initial organometallic platinum layer using a gas injection system (GIS) for 10 sec followed by 20 sec sputtering coating to add an additional inorganic platinum layer. Samples were tilted to a milling angle of 8o. Milling was done iteratively in 3 steps (1: 1 nA–5 μm, 2: 0.5 nA–3 μm, 3: 0.3 nA–1.3 μm) before polishing at 30–50 pA to achieve a final lamella thickness of 100–200 nm. After polishing the EM grids were sputtered again for 2–3 sec.

For data acquisition, 12 (YPD conditions) and 14 (YPG conditions) tilt series were acquired for the wild type experiments and 12 (YPD conditions) and 16 (YPG conditions) tilt series of the *dnm1Δ* sample. Tilt series were collected either on a Titan Krios electron microscope (Thermo Fisher Scientific) equipped with a K2 summit electron detector (Gatan) and a Quantum energy filter using a pixel size of 2.682 Å/pixel or, on a Titan Krios G4 electron microscope (Thermo Fisher Scientific) equipped with Selectris X imaging filter and Falcon 4 direct electron detector using a pixel size of 3.037 Å/pixel. Movies were collected in a tilt range of +68° / −52° starting at +8° pretilt with a 2° tilt increment. Target defocus was set to −2.5 to −4.5 μm. A total electron dose of ~120 e/Å2 per tilt series was used following a dose-symmetric tilt scheme in SerialEM[61,62].

For tomogram reconstruction and volume segmentation, movies of individual projections were motion-corrected and aligned within SerialEM[61]. Combined and dose-filtered stacks were aligned and reconstructed using AreTomo[63]. To improve visualization reconstructed tomograms were binned by 4 and deconvoluted using tom_deconv[64]. Volume segmentation was done manually within the Amira software package (Thermo Fisher Scientific). Final images were prepared using ChimeraX[65].

## Cryo-ET image quantification

Generating the graphs in Figs. 1g, h and 2f–i from Cryo-ET images was done as follows. To measure the diameter of mitochondria, an average of 3 random lines drawn between both ends of OM was considered for the final value. To quantify the ring and constriction of mitochondria, we drew a line along the OM through all sections of a tomogram. This allowed obtaining 2-dimention projections of tomograms which were used to perform the quantification of rings and constriction as described in the TEM image quantification section above. Tomograms also allowed counting the number of cristae. As a first step, a given tomogram was scanned from bottom to top to appreciate the appearing cristae visually. To measure the length of the cristae, we drew a line through the cristae along the tomogram and took the average to 3 longest cristae for the final value. To quantify the cristae crossing the mitochondria, we scanned the tomogram and counted

the cristae which intersect the entire mitochondrial tubule. To quantify the presence of ER at constriction site, we counted those constriction site in which ER was present in close proximity (< 75 nm) per tomogram. To perform comparison with SIM images, we took the projection of four SIM images plane (≈ 300 nm, equivalent to Cryo-ET images) and cropped the same dimension for field of view. To measure the diameter of mitochondria, we drew 3 lines across mitochondria tubules, plotted the line profile and measured the distance between the two peaks. Counting of rings and constriction were done as described in the TEM image quantification section above.

## Fluorescent puncta quantification

GFP puncta quantification of Dnm1, Mmm1 and LacI used in Figs. 3a, c and 6f-g were achieved on a single plane of SIM images using ImageJ Macro. Initially, a plane from the stack where major portions of mitochondrial networks were detected was selected. The background in both channels was then subtracted. This was done by delimiting a region of empty space in the Field of View (FoV) to measure the mean intensity from both channels in this region and subtract it from respective channels. To quantify the puncta, two duplicate images were created and a Gaussian blur of 5 and 7 pixels (the size of the blur depends on the size of the puncta) was introduced. After subtracting the two images, a threshold was applied manually to generate a binary image. Mitochondria were subsequently quantified by using Gaussian blur followed by manual thresholding. We selected the individual yeast cells in the FoV and created a Region of interest (ROI) around each cell, making sure that each ROI contained a single cell and avoiding overlap between two ROIs. The number of puncta in each ROI was counted to finally represent the number of puncta per cell. In parallel, the puncta density on mitochondria was measured. For this purpose, the area of mitochondria in each ROI was evaluated to obtain the ratio of puncta count relative to the area of mitochondria. All the parameters were kept constant for each condition while analyzing the images. This pipeline allowed performing a semi-automated quantification.

Yme2-GFP puncta related to Supplementary Fig. 10h, i were quantified on a maximum intensity projected SIM stack of 5 slices as low signal-to-noise ratio of GFP intensity in single slice was observed. Following background subtraction (see above), we selected individual yeast cells manually by creating ROI around them such that they did not overlap with neighboring cells. The region was duplicated and channels were separated. The yme2-GFP channel was binarized by applying the maximum filter of 2 pixels followed by MaxEntropy thresholding. Tom70-mcherry channel is binarized by introducing a Gaussian blur of 5 pixels followed by automatic mean thresholding. Area by Tom70 was measured and within this ROI, Yme2-GFP puncta was counted. These measurements allowed us to quantify the area of Tom70 and the count of Yme2-GFP puncta. Further analysis was performed on MS Excel. All the parameters were kept constant for each condition while analyzing the images.

## Mitochondrial fission quantification

Mitochondrial fission events quantification in Figs. 3d, e and Supplementary Fig. 7b was performed as follows. We generated a strain in which Dnm1 is tagged with mcherry and Mmm1 with GFP (MCY2199) in which mitochondria were labelled with Mitotracker (#M22426, Invitrogen). This setup allowed following and quantifying the presence of Dnm1-mCherry and Mmm1-GFP at sites of mitochondrial fission. This quantification was performed on raw time-lapse images that were acquired for 3-minute duration at every 10 sec time interval. Before analysis, the image contrast was adjusted in all three channels but in settings where the mitochondria channel was the only one visible. After identifying a fission event, Dnm1 and Mmm1 were checked for their presence in the proximity at the fission site. The fission events were classified into 4 categories including fission event where (1) both Dnm1 and Mmm1 (2) only Dnm1, (3) only Mmm1, and (4) none are

present. The results were plotted as the percentage of total for each category in three independent experiments.

To generate graphs from Fig. 3f, the localization of Dnm1-GFP puncta on the mitochondrial tubules was visually quantified using single planes from z-stacks. After adjusting the image contrast, puncta were categorized into two groups, one localized randomly on mitochondrial tubule (similar to Fig. 3f top image) and another one where puncta specifically localized on mitochondrial constrictions (similar to Fig. 3f bottom image). We plotted the graph as percentage of total of puncta localized in each group.

## PreCOX4 intensity measurement

The analysis from Supplementary Fig. 7c was performed using an ImageJ Macro on single planes from SIM images with the MCY1607 and MCY2161 strains where Tom70 is tagged with GFP and Precox4 with mcherry. Planes from z-stacks where major portions of mitochondrial networks was detected were selected. The background was subtracted as described in the puncta quantification section above. The Tom70-GFP channel was employed to segment mitochondrial networks because of the strong GFP signal. Gaussian blur followed by manual thresholding was used to create a binary image of mitochondria. ROI was created to segment the single cell, considering that ROIs were not overlapping each other. Mean intensity of precox4-mCherry in the mitochondrial ROI was measured. Ultimately, MS Excel and GraphPad prism were used to assemble the data and generate the plots. Each dot in the plots from Supplementary Fig. 7c corresponds to the mean intensity of Precox4-mCherry per cell in one experiment. Bars and error bars respectively represent the mean value and s.d. from three independent experiments.

## Spot assay preparation and quantification

Cultures grown overnight in minimal synthetic media with Dextrose (SD) were pelleted, resuspended at OD600 = 1 per mL of culture, and serially diluted (1:10) three times in water. Five microliters of the dilutions were spotted on SD or minimal synthetic media with Glycerol (SG) plates and grown for 2 to 4 days (Dextrose) or 3 to 6 days (Glycerol) at 23, 30 or 37 °C. "Microarray Profile" plugin for ImageJ was used to quantify each spot in Fig. 5c. Data reported are the mean and s.d. (error bars) from three independent experiments.

## Microfluidic analysis

The microfluidic mold was fabricated using standard soft lithography techniques[66]. To make the chip, we mixed polydimethylsiloxane (PDMS; Sylgard 184) and curing agent in a 10:1 ratio, degassed it with a vacuum pump for 30 min and poured it into the mold. The PDMS was cured by baking at 70 °C for 5 h and then was carefully removed from the mold. A biopsy puncher (1.5 mm, Harris Unicore) was used to create holes for media flow. The surfaces of PDMS and a glass coverslip (24 × 50 mm) were surface-activated using a plasma cleaner (Diener Electronic, Germany) to covalently bond the two elements. For injection of cells into the device, synthetic complete media containing 2% glucose (SD) was filtered using a 0.22-μm polyethersulfone filter (Corning) and loaded into the device using a peristaltic pump (IPCN, Ismatec). Cells from a log-phase culture (0.5 OD600) were gently injected into the device using a 1-ml syringe. A constant media flow (28 μl min⁻¹) was maintained throughout the experiment. Control experiments validating the flow rate, media diffusion into the cavities, nutrient uptake and physiological cell growth were performed as in ref. 66. For experiments with the WT and ADH-DNM1 strains, cells were allowed to divide in SD or SG media.

Cells in the microfluidic device were observed using a fully motorized Axio Observer Z1 inverted microscope (Zeiss), maintaining constant focus with focus stabilization hardware (Definite focus, Zeiss). To reduce phototoxicity, LED light sources were used for phase contrast with specific parameters: 4.0 V–70 ms for phase contrast. Temperature

was kept at 30 °C using a controlled heating unit and an incubation chamber containing the entire microscope setup, including the stage and objectives. Images were taken every 10 min using AxioVision 4 (Zeiss). All image acquisition processes were fully automated and controlled, including temperature, focus, stage position, and time-lapse imaging. Images were captured for over 120 h in standard experiments. Custom Matlab software was utilized to analyze movies, generate graphs, and assess the average number of generations. Survival curves derived from single-cell analysis of both WT and ADH-DNM1, in SD or SG media, were generated utilizing the Kaplan–Meier estimator.

## Respiratory activity assay

The Oxygraph-2k high resolution respirometry system (Oroboros Instrument) was used to measure oxygen consumption rates of WT, dnm1Δ and ADH-DNM1 cells. Cells were grown to stationary phase in Synthetic Dextrose (SD) media and subcultured in SD/SE to reach an optical density at 600 nm of 0.8–1.0. Notably, respiratory growth in SE (2% Ethanol) was preferred to respiratory growth in SG (2% Glycerol) because the Oxygraph tuned out to be non-compatible with cells grown in SG. Measurements were done with $2 \times 10^6$ cells grown at 30 °C. Oligomycin (1 μM) and carbonyl cyanide m-chlorophenyl hydrazone (CCCP, 10 μM) were used to evaluate the proton leakage and maximal respiration (E). To compare the mitochondrial function of yeast strains with different basal respiration (R), the oxygen consumption rates were converted to comparable values by calculating (E-R)/E. The data were analyzed using the DatLab software provided by Oroboros.

## Hoechst PAINT

Hoechst-PAINT acquisition was performed on yeast spheroplasts. To prepare spheroplasts, cells were grown at mid log phase (0.6–0.8 OD600). 1 mL of 3.7% formaldehyde was added to the 10 mL of cell culture and incubated for 20 minutes at 30°C. Cells were then washed with PBS twice, then re-suspended into 3 mL of spheroplast buffer containing 50 mM β-mercaptoethanol (#63689, Sigma-Aldrich) + Zymolyase (#ZE1005, Zymo Research; Orange, CA) at 3 μL/10 OD of culture in PBS. Re-suspended cells were incubated for 30 minutes at 30°C in a 15 mL tube placed horizontally in a rotating incubator. After two washes with PBS, cells were permeabilized using 0.5% of Triton™ X-100 (#T8787, Sigma-Aldrich) in PBS for 30 minutes on a shaker. Finally, spheroplasts were washed 3 times with PBS and stored in 300 μl of PBS at 4°C. Spheroplasts were stable for 2 weeks for further processing and imaging. To immobilize spheroplasts, 25 mm coverslip was coated with 1 mg/ml Concanavalin (#C5275, Sigma-Aldrich) in PBS for 2 h, then washed 3 times with PBS, dried overnight and stored at 4°C in PBS. 30 μL of spheroplasts suspension was mixed with 270 μL of PBS and spread over 25 mm coverslip for 1 hour and then washed 3 times with PBS. The coverslip was placed into Chamlide magnetic chambers and 200 pM of JF646-Hoechst, novel fluorogenic DNA stain (Gift from Janelia farm) conjugated with Janelia Fluor® 646 dye in 1 mL of PBS was added to the chamber. Image acquisition was performed on the same system used for PALM imaging. A 647-nm wavelength laser with 500 mW (at 60 %) was used to acquire 100,000 images without interval with an exposure time of 100 ms. EM gain of the EMCCD was set to 300.

## Hoechst-PAINT data quantification

To quantify information from PAINT data in Supplementary Fig. 10e, f, the.CSV file generated by ThunderSTORM was used and displayed as a Histogram. Even though we used a strain in which Matrix is tagged with mCherry (MCY1949), the mCherry signal was not strong enough to allow automatic segmentation of mitochondria. Consequently, the ROI corresponding to mitochondria in each cell was manually drawn, to measure the area and number of localizations from mitochondrial DNA, which allowed calculating the absolute and density of localizations in each cell.

## Statistical testing

All statistical analyses and graphing were performed using GraphPad Prism 9 or Microsoft Excel. As described in the individual figure legends, Unpaired t-tests (two-tailed), Mann whitney test (two-tailed) or factorial ANOVAs (one- or two-way, corrected for multiple comparisons by Tukey's multiple-comparisons test) were applied to the raw data. A $p$-value $< 0.05$ was considered significant and indicated as follows: ns, not significant, $*p < 0.05$, $**p < 0.01$, $***p < 0.001$, $****p < 0.0001$.

## Reporting summary

Further information on research design is available in the Nature Portfolio Reporting Summary linked to this article.

## Data availability

All the data that support this study are available in the main text or the supplementary materials. Macros used in this paper are accessible at https://github.com/mformanu9/Ringo-project. Source data are provided with this paper.

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

## Acknowledgements

We are grateful to Christof Osman for providing the LacO-LacI strain and to Fanny Pilot-Storck for lending us the Oroboros instrument. We also thank the EM facility of the Max Planck Institute of Biophysics (Frankfurt, Germany), in particular Werner Kühlbrandt for critical reading of the manuscript and Sonja Welsch, Oezkan Yildiz and Juan Castillo for computing support. We thank Nadège Cayet for her help to section embedded samples and TEM acquisition. Research in the Cohen and Teixeira laboratories were supported by the Agence Nationale de la Recherche (ANR) grants, labex DYNAMO (ANR-11-LABX-0011-DYNAMO to MMC and MTT), MOMIT (ANR-17-CE13-0026-01 to MMC) and MITO-FUSION (ANR-19-CE11-0018 to MMC), the "Fondation de la Recherche Medicale" ("Equipe FRM EQU202003010428" to MTT), and The French National Cancer Institute (INCa_15192 to MTT). This project has also received funding from the European Union's Horizon 2020 research and innovation programme under the Marie Skłodowska-Curie (grant agreement N° 101034407 to M.A.M.P). CA was supported by a three-year doctoral grant from the PSL Idex Program (ANR-10-IDEX-0001-02 PSL) and by a fourth-year doctoral grant from the Fondation pour la Recherche Médicale (FRM-FDT202012010377). CZ and AM also acknowledge funding by ANR (ANR-17-CE13-0026-01 to CZ, ANR-21-CE45-003-02 to CZ, ANR-16-CONV-0005 to CZ and ANR-10-LABX-62-IBEID to AM) and by Institut Pasteur. We gratefully acknowledge the Imagopole—Citech of Institut Pasteur (Paris, France) as well as the France–BioImaging infrastructure network supported by the French National Research Agency (ANR-10–INSB–04; Investments for the Future to CZ and AM) for the use of the Zeiss LSM 780 Elyra PS1 microscope and the electron microscopy equipment. We also thank the Max Planck Society and Deutsche Forschungsgesellschaft (FOR2848 to LD).

## Author contributions

M.K.S. acquired and analyzed all SIM, PALM and PAINT data. L.C., M.A.M.P. and N.B.T. prepared all plasmids and strains and performed all western blots and serial dilution assays. M.L. brought assistance to M.K.S. in the PALM and PAINT data acquisition. P.B. and A.M. acquired all TEM data. L.D. and C.K. acquired all Cryo-ET data. M.A.M.P. acquired and analyzed the Respiratory activity data. A.B. and M.T.T. acquired and analyzed the microfluidic data. M.K.S. analyzed all TEM and Cryo-ET data. C.A. and M.K.S. prepared the figures. M.K.S. and M.M.C. conceptualized the project. M.M.C. and C.Z. supervised M.K.S. M.M.C. and M.K.S. wrote the manuscript with inputs from C.Z. and L.D., and all authors reviewed and edited the paper.

## Competing interests

The authors declare no competing interests.
