## [Peer Review file · Nature Communications]

A constricted mitochondrial morphology formed during respiration.

Corresponding Author: Dr Mickael Cohen

Version 0:

Reviewer comments:

Reviewer #1

(Remarks to the Author)

The authors have addressed the concerns raised in the initial submission. In particular, the generation of the ADH1-DNM1 strain has greatly improved the interpretation of the Ringo and HFR morphologies observed and the addition of the LacO-LacI-GFP mtDNA analysis demonstrates the importance of Dnm1 and respiration state for the mtDNA distributions observed. I just have a minor point for text/figure improvement

-in Fig 2j, the large arrowheads obscure the image and labels- these should be adjusted.

Reviewer #2

(Remarks to the Author)

In the manuscript "A constricted mitochondrial morphology optimizes respiration," Singh et al. describe a constricted morphology for mitochondria under different respiratory conditions and correlate its presence to structural and functional outcomes.

We find that the authors addressed some of the specific concerns raised by all reviewers. The controlled expression of intermediate levels of DNM1 allows for more nuanced interpretation of its role. The new mtDNA images collected with the LacO-LacI system are much improved, and of suitable quality for publication.

However, the authors did not address the core issue with the original manuscript, which is their use of correlative observations to infer causality:

Their main argument for the function of the "Ringo" morphology relies on correlations between its prevalence with changes in cellular functions in different nutrient conditions. These functions could instead be a direct consequence of DNM1 levels, since that is their only means of modulating "Ringo."

In my view, there is no way to prove causality without a DNM1-independent manner of inducing "Ringo."

Specific comments:

- line 28: fusion and fission do not regulate ALL mitochondrial functions.
- line 89: the authors now include citations to publications showing multiple regular constrictions of mitochondria as we requested. However, they state Ringo to be unrelated to those, without justification.

• Section "Dnm1, Mdv1 and Fis1 are required for formation of the Ringo morphology" - line 149

The authors report that the deletion of key mitochondrial dynamics proteins leads to a heavily disrupted network, which they term HyperFused Respiratory (HFR). This is not evidence that the deleted proteins are required for Ringo morphology, since all mitochondrial homeostasis appears to be lost.

• Section "Involvement of the Ringo morphology in maintenance of mitochondrial DNA homeostasis during respiration" - line 291

The authors replaced the unclear mtDNA imaging with the more robust LacO-LacI imaging method, which is appreciated, and the data does show effects of SG growth and ADH1-DNM1 expression in mtDNA copy number and the expression of Cox1 and Cox2. The conclusions drawn from the data, however, suffer from heavy overinterpretation. The data does not

specifically involve the Ringo morphology in these changes, or support the causal conclusions of maintenance of mtDNA homeostasis regulating OXPHOS expression.

Reviewer #3

(Remarks to the Author)

In this revised manuscript, Singh et al perform additional characterization of mitochondrial morphology of cells grown in different respiratory conditions and a more thorough analysis of cells with limited mitochondrial constriction by utilizing a repressed allele of Dnm1 to block mitochondrial constriction while not affecting mitochondrial fission. The authors also utilize a more conventional approach than DNA paint to assess mtDNA content and distribution in this Dnm1 hypomorphic strain. While the manuscript is improved over the previous iteration, I still have several substantial concerns regarding the experimental approaches taken by the authors, and find that the conclusions are not fully supported with the experimental evidence provided. As it stands, the authors fall short of demonstrating, or presenting a cohesive model, of how mitochondrial constrictions could promote increased mitochondrial respiration under respiratory conditions.

Specific points:

1. I remain unconvinced that respiring cells have an increased frequency of mitochondrial constriction than cells grown in glycolytic conditions. The authors did not address a primary concern from the initial submission that the increase in mitochondrial mass during glycolysis is not accounted for in their analysis. The 3D reconstructions now included in Video 1 clearly show more complicated mitochondrial networks under respiring conditions, as expected, but do not clearly show elevated levels of constrictions. The quantitative evidence of increased constrictions (Fig. 1g) is displayed on a per cell basis. However, EM tomographs clearly have fewer mitochondria per image to quantify and do not account for the 3D shape. 3D cryo-tomography is a better tool for this analysis, but again is not normalized to mitochondrial length. Single plane images shown by the authors (Fig. 1a, 1b, 1c, 2a, 3a) are misleading because they show "rings" of mitochondria, but are not indicative of their overall tubular and connected nature. Even the images the authors cite comparing mitochondrial morphology in glucose versus glycerol by fluorescence microscopy (ie Egner et al PNAS 2002) make it abundantly clear that mitochondria from respiratory-growing yeast are not drastically hyper-constricted compared to glucose-grown mitochondria.
2. However, even assuming the premise that non-fermentative mitochondria are hyper-constricted, the authors still take a very limited approach to correlate the presence of constriction with increased respiration. They utilize a Dnm1 hypomorph, which they claim because of its reduced expression in respiratory conditions, uncouples mitochondrial morphology from mitochondrial constriction. However, the authors discount the possibility of pleiotropic effects of reduced Dnm1 expression. For example, the Lackner group recently showed that mitochondrial morphology appears normal, though has reduced rates of fission AND fusion in Dnm1 hypomorphic strains (Wisniewski and Lackner, biorxiv, 2024). These reduced dynamics may lead to reduced content mixing and other unintended consequences that could influence the interpretation of their experiments. The authors still fail to use any other orthogonal approach besides the Dnm1 hypomorphic strain to support their conclusions. One experiment they may consider is comparing dnm1-delta cells to atg44-delta cells, which were recently demonstrated to prevent mitochondrial fission without preventing Dnm1-mediated mitochondrial constriction (Fukuda et al Mol Cell 2023; Connor et al JCB 2023; Kurukawa et al Autophagy 2024).
3. It is still not entirely clear the model of how altered mitochondrial leads to improved respiratory function. The authors propose that constrictions would "favor stoichiometric assembly of OXPHOS complexes" but it is not apparent how this might occur. The authors now make mention of the increased density of cristae that occurs in respiration, but still do not address how constrictions of mitochondria may influence inner membrane behavior and lead to improved respiration.

Version 1:

Reviewer comments:

Reviewer #2

(Remarks to the Author)

We have considered the revised version of "A constricted mitochondrial morphology optimizes respiration," as well as the rebuttal letter. Many of our concerns were shared by Reviewer 3. Unfortunately, the authors did not address our main concerns, and we find that the data still do not justify the main conclusion.

Summary of main issues:

1. Causal link between "Ringo" morphology and "optimal" respiration is not supported by data. The authors present no data where "Ringo" is abrogated (while maintaining mitochondrial tubular form), and/or where "Ringo" is induced and tested for predicted increase in respiration (while maintaining other factors such as carbon source). The atg44 control does not achieve this, since it can also "Ringo" and thus is equivalent to a wild-type condition as regards morphology. The authors claim that loss of respiration following DNM1-mediated mitochondrial disruption is proof that "Ringo" is responsible. But many other structural aspects of mitochondria are disrupted beyond "Ringo" in that genetic background, preventing causal interpretation.

2. Causal link between mtDNA homeostasis and “Ringo” is not supported by data. As in point 1, DNM1-mediated fission has already been demonstrated to be required for mtDNA replication and gene expression upregulation during respiratory transition. This does not point to “Ringo” as cause, but only offers correlation.

3. Distinction between previous reports (“CoMIC”, “pearling,” etc) and “Ringo” is not supported by statistics. The authors claim that “Ringo” is different because it is stable, does not lead to fission, and engages the OMM as well as IMM. Yet, they do not offer quantification of lifetimes or fission rates (perhaps fission rates overall are suppressed in conditions where “Ringo” is prevalent). In fact, many of the reports in the literature also mention OMM being engaged in mitochondrial pearling.

In the interest of helping the authors to make the link between “Ringo” and function, we offer a few suggestions. In a recent paper a similar morphology has been reported in respiratory conditions mediated by hexokinase rings (<https://doi.org/10.1016/j.molcel.2024.06.009>). The removal of hexokinases may be able to abrogate “Ringo” without disruption of mitochondrial ultrastructure. Calcium uptake and actin polymerization have also been linked to regular constrictions of mitochondria (<https://doi.org/10.1038/ncomms15754>, <https://doi.org/10.1083/jcb.201709111>). Thus, blocking or inducing those factors may modulate “Ringo” and provide the necessary experimental conditions to make rigorous conclusions about the direct effects of Ringo beyond the pleiotropic consequences of fission modulation.

Reviewer #3

(Remarks to the Author)

My previous review of the manuscript centered on two major issues:

1. Skepticism that mitochondria under glycolytic conditions are constricted to a degree not previously reported in the literature.
2. The lack of causality and understanding of the connection between ringo and increased respiratory activity.

Regarding point 1, I appreciate the authors feel that they have shown a substantial amount of evidence in favor of their claims. Though I remain highly doubtful, I do not want my continued skepticism of this point to stand in the way of publication of the findings.

Regarding point 2, the added data on Atg44 support the authors claim that Dnm1 and Atg44 are differentially required for respiratory growth. However, these data are in direct conflict with published observations of Fukuda et al (see their Fig. 1C) and Connor et al (see their Fig.1E), and even if true, may be explained by reasons other than ringo (ie, the differential requirement of Dnm1 and Atg44 in mitophagy). The authors are also highly dismissive of the point that altering Dnm1 levels by adjusting its promoter may have unintended consequences, going as far as to question the ability of Lackner and colleagues to quantify fusion/fission events by confocal microscopy, a wholly unfounded claim. Ultimately, I agree with the point of Reviewer 2 “there is no way to prove causality without a DNM1-independent manner of inducing ringo,” which has still not been satisfyingly addressed by the authors. I cannot support publication of this manuscript without further substantiation of their claims that ringo morphology has any bearing on mitochondrial respiration.

Reviewer #4

(Remarks to the Author)

REVIEWER COMMENTS

Reviewer #1 (Remarks to the Author):

The authors have addressed the concerns raised in the initial submission. In particular, the generation of the ADH1-DNM1 strain has greatly improved the interpretation of the Ringo and HFR morphologies observed and the addition of the LacO-LacI-GFP mtDNA analysis demonstrates the importance of Dnm1 and respiration state for the mtDNA distributions observed.

I just have a minor point for text/figure improvement

-in Fig 2j, the large arrowheads obscure the image and labels- these should be adjusted.

Response R1.1: We truly thank Reviewer 1 for stating that we have addressed the concerns raised in the initial submission. We agree that the arrowheads in Fig. 2j and Supplementary Fig. 6f were too large. This problem has been fixed with smaller arrowheads in this revised submission.

Reviewer #2 (Remarks to the Author):

In the manuscript "A constricted mitochondrial morphology optimizes respiration," Singh et al. describe a constricted morphology for mitochondria under different respiratory conditions and correlate its presence to structural and functional outcomes.

We find that the authors addressed some of the specific concerns raised by all reviewers. The controlled expression of intermediate levels of DNM1 allows for more nuanced interpretation of its role. The new mtDNA images collected with the Laco-LacI system are much improved, and of suitable quality for publication.

However, the authors did not address the core issue with the original manuscript, which is their use of correlative observations to infer causality:

Their main argument for the function of the "Ringo" morphology relies on correlations between its prevalence with changes in cellular functions in different nutrient conditions. These functions could instead be a direct consequence of DNM1 levels, since that is their only means of modulating "Ringo."

In my view, there is no way to prove causality without a DNM1-independent manner of inducing "Ringo."

Response R2.1: We thank the reviewer for the positive comments and acknowledge the criticism on causality which we do not feel justified as explained in Response R2.5.

Specific comments:

- line 28: fusion and fission do not regulate ALL mitochondrial functions.

Response R2.2: Agreed. ALL has been replaced by MOST. Page 1, line 28.

- line 89: the authors now include citations to publications showing multiple regular constrictions of mitochondria as we requested. However, they state Ringo to be unrelated to those, without justification.

Response R2.3: The constrictions of the Ringo morphology are stable and not transient. They do not lead to mitochondrial fission. They are generated independent of contacts with ER membranes. They exclusively take place during respiration. Most importantly, they are shown to take place on outer membranes, while all mentioned citations show constrictions of the matrix. All these distinctions are now clearly exposed in the revised submission:

Page 3, lines 7-12: "Numerous previous studies have reported transient constrictions of the mitochondrial matrix in processes as diverse as actin polymerization at the ER24, mitochondrial depolarization²⁵ or mitochondrial division^{26,27}. However, a respiratory mitochondrial morphology characterized by seemingly regular and stable constrictions of the outer membrane that do not lead to mitochondrial fission has to our knowledge not been reported before. We chose to name this unknown morphology as "Ringo"."

Page 6, lines 13-17: " These results support a mechanism in which Dnm1 triggers repetitive constrictions within the Ringo mitochondrial network that do not result in fission in the absence of contacts with the ER. This further emphasizes the distinction between Ringo and previously known transient constrictions of the mitochondrial matrix that take place at sites of contact with the ER24–27."

- Section "Dnm1, Mdv1 and Fis1 are required for formation of the Ringo morphology" - line 149
The authors report that the deletion of key mitochondrial dynamics proteins leads to a heavily disrupted network, which they term HyperFused Respiratory (HFR). This is not evidence that the deleted proteins are required for Ringo morphology, since all mitochondrial homeostasis appears to be lost.

Response R2.4: Lack of Dnm1, Mdv1 and Fis1 lead to the same HFR mitochondrial morphology as opposed to lack of Caf4 that does not affect the formation of Ringo networks. This data alone show that the Ringo morphology cannot be formed without Dnm1, Mdv1 or Fis1 (Fig. 2a-d). The three proteins are thus required to form Ringo networks. In the revised manuscript, we now show that lack of Atg44, which is as essential as Dnm1, Mdv1 or Fis1 for mitochondrial fission, does not affect the formation of Ringo-like networks (New Fig. 4c-e). In addition, we now include an important data showing that lack of Dnm1, Mdv1 and Fis1 affect respiratory growth as opposed to the lack of Caf4 or Atg44 which maintain normal respiratory growth (New Fig. 4f and Supplementary Fig. 8d). In light of this new set of data and since Dnm1 needs Mdv1 and Fis1 for its recruitment to mitochondria, we maintain that Dnm1, Mdv1 and Fis1 are required to promote formation of Ringo networks.

- Section "Involvement of the Ringo morphology in maintenance of mitochondrial DNA homeostasis during respiration" - line 291

The authors replaced the unclear mtDNA imaging with the more robust LacO-LacI imaging method, which is appreciated, and the data does show effects of SG growth and ADH1-DNM1 expression in mtDNA copy number and the expression of Cox1 and Cox2. The conclusions drawn from the data, however, suffer from heavy overinterpretation. The data does not specifically involve the Ringo morphology in these changes, or support the causal conclusions of maintenance of mtDNA homeostasis regulating OXPHOS expression.

Response R2.5: We thank the reviewer for recognizing the improvements obtained with the LacO-LacI and the ADH-DNM1 systems. However, according to this reviewer, the inhibition of the Ringo morphology formation in the ADH-DNM1 strain is unlikely to be the only defect in this strain,

precluding to conclude that the defects seen on *cox1/2* expression and mtDNA are caused by inhibition of the Ringo morphology formation.

As emphasized in the revised version of the manuscript, during ADH-driven expression of Dnm1 under respiratory growth, Dnm1 is untagged and expressed at the level seen in WT cells during fermentation (Page 7, lines 33-35). In this context, Dnm1 is fully functional. Upon respiration, Dnm1 expressed from the ADH1 promoter should thus carry out all the functions that are carried out by Dnm1 upon fermentation in WT cells. Consistent with this, Hyperfused or HFR networks were not observed in ADH-DNM1 cells during respiration (Fig. 5b; SG), indicating that the main function of Dnm1 in mitochondrial fission is not affected. If mitochondrial fission is not affected, it is very unlikely that other functions of Dnm1 besides formation of Ringo networks would be. This is even more unlikely, since formation of Ringo networks is significantly decreased but not abolished (Fig. 5b; SG), confirming again that Dnm1 is fully functional. We conclude that formation of the Ringo morphology is required for respiration (Page 8, lines 6-11).

Another important point is that the functions mediated by Dnm1 at mitochondria which, if inhibited, could affect (1) mtDNA distribution, (2) resulting in defective stoichiometry of OXPHOS complexes and, as a consequence, (3) decreased respiration, are restricted to mitochondrial fission or formation of Ringo networks. We show that inhibition of mitochondrial fission in *atg44Δ* cells does not impact respiration (New Fig. 4f and Supplementary Fig. 8d) and that mitochondrial fission is not affected in ADH-DNM1 cells (Fig. 5b; SG). In this context, concluding that defects seen on *cox1/2* expression and mtDNA are caused by inhibition of the Ringo morphology formation in ADH-DNM1 cells (Page 9, lines 19-21) is motivated by logic but not by heavy overinterpretation.

Reviewer #3 (Remarks to the Author):

In this revised manuscript, Singh et al perform additional characterization of mitochondrial morphology of cells grown in different respiratory conditions and a more thorough analysis of cells with limited mitochondrial constriction by utilizing a repressed allele of Dnm1 to block mitochondrial constriction while not affecting mitochondrial fission. The authors also utilize a more conventional approach than DNA paint to assess mtDNA content and distribution in this Dnm1 hypomorphic strain. While the manuscript is improved over the previous iteration, I still have several substantial concerns regarding the experimental approaches taken by the authors, and find that the conclusions are not fully supported with the experimental evidence provided. As it stands, the authors fall short of demonstrating, or presenting a cohesive model, of how mitochondrial constrictions could promote increased mitochondrial respiration under respiratory conditions.

Response R3.1: We appreciate that the reviewer mentions that the manuscript is improved. However, we disagree with his/her criticisms as explained in the responses below.

Specific points:

1. I remain unconvinced that respiring cells have an increased frequency of mitochondrial constriction than cells grown in glycolytic conditions. The authors did not address a primary concern from the initial submission that the increase in mitochondrial mass during glycolysis is not accounted for in their analysis. The 3D reconstructions now included in Video 1 clearly show more complicated mitochondrial networks under respiring conditions, as expected, but do not clearly show elevated

levels of constrictions. The quantitative evidence of increased constrictions (Fig. 1g) is displayed on a per cell basis. However, EM tomographs clearly have fewer mitochondria per image to quantify and do not account for the 3D shape. 3D cryo-tomography is a better tool for this analysis, but again is not normalized to mitochondrial length. Single plane images shown by the authors (Fig. 1a, 1b, 1c, 2a, 3a) are misleading because they show “rings” of mitochondria, but are not indicative of their overall tubular and connected nature. Even the images the authors cite comparing mitochondrial morphology in glucose versus glycerol by fluorescence microscopy (ie Egner et al PNAS 2002) make it abundantly clear that mitochondria from respiratory-growing yeast are not drastically hyper-constricted compared to glucose-grown mitochondria.

Response R3.2: The reviewer is reluctant to believe that mitochondrial networks undergo massive constrictions during respiration despite all the work we already achieved to demonstrate this. We feel totally sorry about this but emphasize 3 distinct lines of evidence that further comfort our claim:

(1) We point out that besides SIM data in Fig. 1a, 1b, 1c, 2a and 3a, we provide EM (Fig. 1d and Supplementary Fig. 4a) and Cryo-ET (Fig. 1e) data that confirm these constrictions. To consolidate the quantification in Fig. 1g, we now show that constrictions per μM length of mitochondrial networks increase by 4-fold in respiration as compared to fermentation (New Supplementary Fig. 4b).

(2) Careful observation of Fig. 4b in Egner et al allows to discern these increased constrictions but certainly not as obviously than when using SIM. Yet, consistent with these increased constrictions during respiration, Visser et al. in *S. cerevisiae* (1995) as well as Zheng et al. in *S. pombe* (2019) have observed fragmented mitochondrial matrices at lower resolution than SIM.

(3) Finally, we invite the reviewers to refer to response R3.3 and our new data on Atg44 (New Fig. 4c-f and new supplementary Fig. 8d). If Reviewer 3 believes that mitochondrial networks from *atg44 Δ* cells are hyper-constricted during respiratory growth in Fig. 3 from Furukawa et al. (2024) and in our new Fig. 4c, it should then become easier to accept our demonstration that Ringo networks from Fig. 1a, 1b, 1c, 2a and 3a are hyperconstricted.

2. However, even assuming the premise that non-fermentative mitochondria are hyper-constricted, the authors still take a very limited approach to correlate the presence of constriction with increased respiration. They utilize a Dnm1 hypomorph, which they claim because of its reduced expression in respiratory conditions, uncouples mitochondrial morphology from mitochondrial constriction. However, the authors discount the possibility of pleiotropic effects of reduced Dnm1 expression.

Response R3.3: The expression of endogenous Dnm1 from WT cells increases by two-fold in respiratory as compared to fermentative growth (Fig. 3a-b). In the ADH-DNM1 strain, endogenous Dnm1 is not tagged and its expression in respiratory growth decreases by two-fold as compared to WT cells in respiratory growth (Fig. 5a). Endogenous Dnm1 expressed from the ADH1 promoter in respiratory growth is thus expressed at a level comparable to endogenous Dnm1 expressed from WT cells in fermentative growth. This is now clearly explained in the revised manuscript (Page 7, lines 33-35). In this context, Dnm1 expressed from the ADH1 promoter is fully functional as testified by its capacity to maintain efficient mitochondrial fission in respiratory conditions because of the absence of Hyperfused or HFR networks but also to promote formation of Ringo networks together with tubular networks (Fig. 5b).

For example, the Lackner group recently showed that mitochondrial morphology appears normal, though has reduced rates of fission AND fusion in Dnm1 hypomorphic strains (Wisniewski and Lackner, biorxiv, 2024). These reduced dynamics may lead to reduced content mixing and other unintended consequences that could influence the interpretation of their experiments. The authors

still fail to use any other orthogonal approach besides the Dnm1 hypomorphic strain to support their conclusions.

Response R3.4: The study by Wisniewski and Lackner is distinct from ours in several regards: 1. They use confocal microscopy and not SIM. This questions their capacity to precisely quantify fusion/fission events, in aggregated mitochondrial networks especially; 2. All their analysis are performed in fermentative conditions. They do not assess fusion/fission in respiratory settings. 3. In their Dnm1 hypomorphic strains, Dnm1 is C-terminally tagged which affects its intrinsic activity, not its expression. In this regard, the ADH-DNM1 strain we use is by no mean comparable to the Dnm1 hypomorphic strains they use as explained above (Response R3.3).

One experiment they may consider is comparing *dnm1*-delta cells to *atg44*-delta cells, which were recently demonstrated to prevent mitochondrial fission without preventing Dnm1-mediated mitochondrial constriction (Fukuda et al Mol Cell 2023; Connor et al JCB 2023; Kurukawa et al Autophagy 2024).

Response R3.5: We considered analyzing *atg44*Δ cells just before receiving the comments of the reviewers. Our results are compiled in New Fig. 4c-f. We confirmed that under fermentation, cells lacking Atg44 are hyperfused similar to cells lacking Dnm1. However, under respiration we found that cells lacking Atg44 do not form HFR networks but Ringo-like mitochondrial morphologies. Consistent with this, *atg44*Δ cells do not display respiratory growth defects, as opposed to *dnm1*Δ, *mdv1*Δ and *fis1*Δ cells (New Fig. 4f and Supplementary Fig. 8d). These new results confirm that respiratory growth defects are unlikely caused by inhibition of mitochondrial fission but by an alternative function of Dnm1 that requires its recruitment to mitochondria by Mdv1 and Fis1. Our results with the ADH-DNM1 strain indicate that this function required for respiration consists in the formation of Ringo networks.

Last but not least, Ringo-like networks seen in *atg44*Δ cells during respiratory growth were previously interpreted as hyper constricted networks by Dnm1 due to the loss of inner membrane fission by Atg44 (Kurukawa et al. Autophagy 2024). This is in total agreement with our interpretation of Ringo networks. Yet, we now show that the respiratory morphology seen in the absence of Atg44 is similar to the respiratory morphology seen in the presence of Atg44. In other words, Atg44 is not involved in the formation of Ringo networks.

3. It is still not entirely clear the model of how altered mitochondrial leads to improved respiratory function. The authors propose that constrictions would “favor stoichiometric assembly of OXPHOS complexes” but it is not apparent how this might occur. The authors now make mention of the increased density of cristae that occurs in respiration, but still do not address how constrictions of mitochondria may influence inner membrane behavior and lead to improved respiration.

Response R3.6: Osman and colleagues have previously shown that each mtDNA copy has a “sphere of influence” and remains spatially linked to its gene products (ref). This point was mentioned in the results section of the previous submission. We now moved this important consideration in the discussion section to expose how the Ringo morphology could favor stoichiometric assembly of OXPHOS complexes: “Ringo natural constrictions independent of ER membranes regulate mtDNA distribution during respiration, which is in line with the Ringo requirement in synthesis of OXPHOS subunits that are specifically encoded by mtDNA (Fig. 6a-d). Consistent with this, it has been shown that each mtDNA copy has a “sphere of influence” and remains spatially linked to its gene products (Jakubke et al. Sci Adv 2021). We conclude that the Ringo morphology contributes to respiration by

promoting accurate distribution of mt nucleoids along the mitochondrial network thereby favoring optimal assembly of OXPHOS complexes during respiratory growth” (Page 9, lines 35-41).

REVIEWER COMMENTS

Reviewer #2 (Remarks to the Author):

We have considered the revised version of "A constricted mitochondrial morphology optimizes respiration," as well as the rebuttal letter. Many of our concerns were shared by Reviewer 3. Unfortunately, the authors did not address our main concerns, and we find that the data still do not justify the main conclusion.

Summary of main issues:

1. Causal link between “Ringo” morphology and “optimal” respiration is not supported by data. The authors present no data where “Ringo” is abrogated (while maintaining mitochondrial tubular form), and/or where “Ringo” is induced and tested for predicted increase in respiration (while maintaining other factors such as carbon source). The *atg44* control does not achieve this, since it can also “Ringo” and thus is equivalent to a wild-type condition as regards morphology. The authors claim that loss of respiration following DNM1-mediated mitochondrial disruption is proof that “Ringo” is responsible. But many other structural aspects of mitochondria are disrupted beyond “Ringo” in that genetic background, preventing causal interpretation.

Response R2.1: Unfortunately, the reviewers make abstraction of key data and text in the manuscript. We do present data where Ringo is abrogated while maintaining mitochondrial tubular form in Fig. 5b with the ADH-DNM1 system. We do present data where Ringo is induced and tested for predicted increase in respiration while maintaining other factors such as carbon source in Fig. 5e. We never claim that DNM1 disruption is proof that Ringo is responsible for loss of respiration but write the opposite: “Absence of Dnm1 does... not allow distinguishing on whether the defective respiratory growth phenotype of *dnm1Δ* cells correlates with abrogation of the Ringo morphology or with the abrogation of mitochondrial fission” (Page 6, lines 38-40), which is similar to saying that many other structural aspects of mitochondria are disrupted beyond Ringo in that genetic background, preventing causal interpretation. The role of the *Atg44* control is not intended to obtain this causal aspect but to provide additional evidence that inhibiting mitochondrial fission does not affect respiratory growth as opposed to inhibiting Ringo formation through ablation of DNM1. Most importantly, it is by blocking increase of Dnm1 expression with the ADH-DNM1 system (of which Reviewers 2 and 4 make abstraction) that comfort the hypothesis that Ringo may favor respiration. In the ADH-DNM1 strain (see also Response R3.3), endogenous Dnm1 is not tagged and expressed at a level comparable to endogenous Dnm1 expressed from WT cells in fermentative growth. Dnm1 expressed from the ADH1 promoter is fully functional as testified by its capacity to maintain efficient mitochondrial fission in respiratory conditions.

At this point, and despite precisions provided above, we chose to comply with the reviewers request not to claim that the Ringo morphology optimizes respiration. We thus revised the whole manuscript to remove any causal relationship between Ringo and respiration and only highlight correlations between inhibition of Ringo and decreased respiration (see manuscript with track-changes and response R2.4).

2. Causal link between mtDNA homeostasis and “Ringo” is not supported by data. As in point 1, DNM1-mediated fission has already been demonstrated to be required for mtDNA replication and gene expression upregulation during respiratory transition. This does not point to “Ringo” as cause, but only offers correlation.

Response R2.2: We have already answered to this comment in the previous round of review with factual points. We are aware and our paper does acknowledge that DNM1-mediated fission has already been demonstrated to be required for mtDNA replication (Page 11, lines 1-2). But we never write that this points to Ringo as a cause for respiration and mtDNA distribution.

Similar to respiration, we have now revised the whole manuscript to remove any causal relationship between Ringo and mt-DNA distribution and only highlight correlations between inhibition of Ringo and perturbed mtDNA distribution (see manuscript with track-changes and response R2.4).

3. Distinction between previous reports (“CoMIC”, “pearling,” etc) and “Ringo” is not supported by statistics. The authors claim that “Ringo” is different because it is stable, does not lead to fission, and engages the OMM as well as IMM. Yet, they do not offer quantification of lifetimes or fission rates (perhaps fission rates overall are suppressed in conditions where “Ringo” is prevalent). In fact, many of the reports in the literature also mention OMM being engaged in mitochondrial pearling.

Response R2.3: Constriction of Mitochondrial Inner Compartments (CoMIC) occur at ER–mitochondria contact sites and are initiated independently of Drp1 action and OMM constriction. The large majority of reports that mention OMM being engaged in mitochondrial pearling deal with constrictions generated prior to mitochondrial fission. To our knowledge, the only manuscript reporting OMM constrictions that are not involved in mitochondrial fission are HK1-rings that do not depend on DRP1 but on the ER and inhibit mitochondrial fission. On the other hand, Ringo constrictions are mediated by Dnm1 independently of ER membranes and do not lead to mitochondrial fission. Yet and again, reviewers make abstraction of our data as we do show that formation of Ringo networks does not inhibit mitochondrial fission. Mitochondrial fission over 180 seconds is equivalent in tubular and Ringo networks with 0.279 ± 0.9 event per Ringo network and 0.282 ± 0.5 event per tubular network (Supplementary Fig.7b).

In the revised manuscript, we have extensively revisited the discussion to highlight the difference between CoMIC, HK1-rings and Ringo networks with numerous statistics. These statistics were present in figure panels from the beginning but have now been compiled in the main text (Page 9; line 31 to Page 10; line 33). This new section confirms that Ringo is distinct from CoMIC and HK1-Rings.

In the interest of helping the authors to make the link between “Ringo” and function, we offer a few suggestions.

In a recent paper a similar morphology has been reported in respiratory conditions mediated by hexokinase rings (<https://doi.org/10.1016/j.molcel.2024.06.009>). The removal of hexokinases may be able to abrogate “Ringo” without disruption of mitochondrial ultrastructure. Calcium uptake and actin polymerization have also been linked to regular constrictions of mitochondria (<https://doi.org/10.1038/ncomms15754>, <https://doi.org/10.1083/jcb.201709111>). Thus, blocking or inducing those factors may modulate “Ringo” and provide the necessary experimental conditions to make rigorous conclusions about the direct effects of Ringo beyond the pleiotropic consequences of fission modulation.

Response R2.4: We thank the reviewers with attempting to help us improving the demonstration that Ringo is involved in respiration and maintenance of mtDNA distribution. However, their suggestions assimilate mammalian cells to yeast cells. Applying mammalian concepts into yeast is not as straightforward as it seems. Moreover, Ringo is radically distinct from CoMIC and HK1-Rings, only taking

into consideration that both systems take place at mito-ER contact sites whereas Ringo does not. We have highlighted the differences between CoMIC, HK1-Rings and Ringo in the revised manuscript (see response R2.3). At this point, our mechanistic investigation on Ringo formation involves increased expression of Dnm1 together with a role of Mdv1 and Fis1 as well as a non-involvement of Atg44 or ER contacts. More work is required to understand the mitochondrial recruitment of Dnm1 away from sites of contact with the ER, which may provide additional means to confirm the role of Ringo networks in respiration. Nonetheless, our data with Atg44 and the ADH-DNM1 system already provide hints toward a model in which Ringo favors respiration through proper distribution of mt nucleoids, regardless of mitochondrial fission. Reviewer 1 agreed with our view but Reviewers 2 and 3 do not.

In consequence, we revised the title, the abstract, the main text and the discussion to remove any claim that Ringo networks are required for optimal respiration (see manuscript with track-changes). We hope this will avoid further delay in publishing this work.

Reviewer #3 (Remarks to the Author):

My previous review of the manuscript centered on two major issues:

1. Skepticism that mitochondria under glycolytic conditions are constricted to a degree not previously reported in the literature.
2. The lack of causality and understanding of the connection between ringo and increased respiratory activity.

Regarding point 1, I appreciate the authors feel that they have shown a substantial amount of evidence in favor of their claims. Though I remain highly doubtful, I do not want my continued skepticism of this point to stand in the way of publication of the findings.

Response R3.1: We thank the reviewer not to stand in the way of publication of our findings despite his/her skepticism that mitochondria under respiratory conditions are constricted to a degree not previously reported in the literature. However, we still do not understand this skepticism given our results and the straightforward responses we provided in the previous round of review.

Regarding point 2, the added data on Atg44 support the authors claim that Dnm1 and Atg44 are differentially required for respiratory growth. However, these data are in direct conflict with published observations of Fukuda et al (see their Fig. 1C) and Connor et al (see their Fig.1E), and even if true, may be explained by reasons other than ringo (ie, the differential requirement of Dnm1 and Atg44 in mitophagy).

Response R3.2: We appreciate that our data on Atg44/Mdi1 convince the reviewer that Dnm1 and Atg44/Mdi1 are differentially required for respiratory growth. However, these data are not in conflict with Fig. 1E from Connor et al. where colony size of *dnm1Δ* cells are clearly smaller than those of *WT* and *mdi1Δ* cells at 37°C on YPEG media. These results are in perfect agreement with our observation obtained with W303 cells at 30 or 37°C.

As for Fig. 1C of Fukuda et al., the data deals with the demonstration that GFP is no longer cleaved from Idh1-GFP in *Atg44Δ* cells grown in media lacking Nitrogen, demonstrating that Atg44 is required for mitophagy. We did not investigate the formation of Ringo networks upon Nitrogen depletion and

mitophagy induction. We find the demonstration of Atg44 involvement in mitophagy very interesting and convincing and we certainly do not question these results from Fukuda et al.

Regarding the comment that our results may be explained by a differential requirement of Dnm1 and Atg44 in mitophagy, we need to reiterate that we did not investigate the Ringo morphology in mitophagy inducing conditions. This would clearly be interesting in the future but is above the scope of the current study. We emphasize that we restricted our investigation on Atg44 to mitochondrial morphology analysis and respiratory growth in YPD and YPG at the exponential phase where Atg8 and the mitophagy receptor Atg32 are hardly expressed and mitophagy is thus hardly induced. We find that cells lacking Atg44 keep a Ringo-like morphology and are not affected for respiratory growth, just like WT cells. If inhibition of mitophagy in *atg44Δ* would explain our findings, this would mean that mitophagy is also inhibited in WT cells. This can obviously be excluded meaning that if our data are true (and they are), the differential requirement of Dnm1 and Atg44 in mitophagy does not explain our findings.

Instead, Dnm1 participate in the formation of Ringo constrictions whereas Atg44 does not, which also excludes that inhibition of mitochondrial fission in *dnm1Δ* and *atg44Δ* cells can explain the defective respiratory growth in the absence of Dnm1. With all due respect to the reviewer, mitochondrial fission or mitophagy can be excluded but inhibition of Ringo constrictions cannot.

The authors are also highly dismissive of the point that altering Dnm1 levels by adjusting its promoter may have unintended consequences, going as far as to question the ability of Lackner and colleagues to quantify fusion/fission events by confocal microscopy, a wholly unfounded claim.

Response R3.3: We have nothing against Lackner and colleagues and truly appreciate their study showing that partial decrease of fission generates concomitant partial decrease of fusion. Our comment on the difficulty to quantify fusion/fission events by confocal microscopy is just motivated by facts as we experienced this issue in our recent publication from 2024 (Alsayyah et al. Plos Biology) where we quantified fusion and fission events by SIM and confocal microscopy and figured that fission/fusion events in aggregated mitochondria are better resolved by SIM than confocal imaging. That said, we were not dismissive of the point that altering Dnm1 levels by adjusting its promoter may have unintended consequences. The reviewer suggested that ADH-DNM1 was a hypomorph allele to reproach us discounting the possibility of pleiotropic effects of reduced Dnm1 expression. Our response was not dismissive but factual:

“The expression of endogenous Dnm1 from WT cells increases by two-fold in respiratory as compared to fermentative growth (Fig. 3a-b). In the ADH-DNM1 strain, endogenous Dnm1 is not tagged and its expression in respiratory growth decreases by two-fold as compared to WT cells in respiratory growth (Fig. 5a). Endogenous Dnm1 expressed from the ADH1 promoter in respiratory growth is thus expressed at a level comparable to endogenous Dnm1 expressed from WT cells in fermentative growth... In this context, Dnm1 expressed from the ADH1 promoter is fully functional as testified by its capacity to maintain efficient mitochondrial fission in respiratory conditions because of the absence of Hyperfused or HFR networks but also to promote formation of Ringo networks together with tubular networks (Fig. 5b).”

Ultimately, I agree with the point of Reviewer 2 “there is no way to prove causality without a DNM1-independent manner of inducing ringo,” which has still not been satisfyingly addressed

by the authors. I cannot support publication of this manuscript without further substantiation of their claims that ringo morphology has any bearing on mitochondrial respiration.

Response R3.4: As explained to reviewer 2 (please refer to response R2.4), our mechanistic investigation on Ringo formation involves increased expression of Dnm1 together with a role of Mdv1 and Fis1 as well as a non-involvement of Atg44 or ER contacts. We convey that more work is required to understand the mitochondrial recruitment of Dnm1 away from sites of contact with the ER, which may provide additional means to confirm the role of Ringo networks in respiration. Nonetheless, our data with Atg44 and the ADH-DNM1 system already provide hints toward a model in which Ringo favors respiration through proper distribution of mt nucleoids, regardless of mitochondrial fission. Reviewer 1 agreed with our view but Reviewers 2 and 3 do not.

In consequence, we revised the title, the abstract, the main text and the discussion to remove any claim that Ringo networks are required for optimal respiration and only highlight correlations between inhibition of Ringo and decreased respiration or perturbed mtDNA distribution (see manuscript with track-changes). We hope this will avoid further delay in publishing this work.

Reviewer #4 (Remarks to the Author):

Response R4.1: As far as we understand Reviewer 4 works with Reviewer 2. Please refer to our responses to Reviewer 2.